# SNAP-25, but not SNAP-23, is essential for photoreceptor development, survival, and function in mice

Mengjia Huang[1,2], Chun Hin Chow[1,2], Akshay Gurdita[3,4], Hidekiyo Harada[4], Victor Q. B. Pham Truong[3,4], Sarah Eide[2], Hong-Shuo Sun[2,5], Zhong-Ping Feng[2], Philippe P. Monnier[2,4,6], Valerie A. Wallace[3,4,6] & Shuzo Sugita [1,2 ✉]

SNARE-mediated vesicular transport is thought to play roles in photoreceptor glutamate exocytosis and photopigment delivery. However, the functions of Synaptosomal-associated protein (SNAP) isoforms in photoreceptors are unknown. Here, we revisit the expression of SNAP-23 and SNAP-25 and generate photoreceptor-specific knockout mice to investigate their roles. Although we find that *SNAP-23* shows weak mRNA expression in photoreceptors, SNAP-23 removal does not affect retinal morphology or vision. *SNAP-25* mRNA is developmentally regulated and undergoes mRNA trafficking to photoreceptor inner segments at postnatal day 9 (P9). SNAP-25 knockout photoreceptors develop normally until P9 but degenerate by P14 resulting in severe retinal thinning. Photoreceptor loss in SNAP-25 knockout mice is associated with abolished electroretinograms and vision loss. We find mistrafficked photopigments, enlarged synaptic vesicles, and abnormal synaptic ribbons which potentially underlie photoreceptor degeneration. Our results conclude that SNAP-25, but not SNAP-23, mediates photopigment delivery and synaptic functioning required for photoreceptor development, survival, and function.

[1] Division of Experimental & Translational Neuroscience, Krembil Brain Institute, University Health Network, Toronto, ON M5T 0S8, Canada. [2] Department of Physiology, Temerty Faculty of Medicine, University of Toronto, Toronto, ON M5S 1A8, Canada. [3] Department of Laboratory Medicine and Pathobiology, University of Toronto, Toronto, ON M5S 1A8, Canada. [4] Donald K. Johnson Eye Institute, University Health Network, Toronto, ON M5T 0S8, Canada. [5] Department of Anatomy, Temerty Faculty of Medicine, University of Toronto, Toronto, ON M5S 1A8, Canada. [6] Department of Ophthalmology & Vision Sciences, University of Toronto, Toronto, ON M5T 3A9, Canada. ✉email: Shuzo.Sugita@uhnresearch.ca

The retina is a highly organized structure containing light-sensing cells called photoreceptors. Photoreceptor cells have outer segments that are a modified sensory cilium made up of stacked membranous disks containing proteins and photopigments for light transduction[1]. Degeneration and dysfunction of photoreceptor cells underlie various diseases including retinitis pigmentosa, cone–rod dystrophies, and Bardet–Biedl syndrome[2–4]. In Bardet–Biedl syndrome, disrupted photopigment trafficking precedes cell death[3]. Additionally, defective synaptic transmission can be a contributing factor to retinitis pigmentosa and cone–rod dystrophies. Therefore, understanding the mechanisms underlying photoreceptor trafficking processes is very important.

Visual information processing begins from the ciliary end of photoreceptors, which consists of an outer segment and an inner segment. Photoreceptor outer segments are made up of stacked membranous disks containing proteins that are essential for phototransduction such as opsin pigments which are G-protein coupled receptors (GPCRs) bound to 11-cis-retinal. Cone photoreceptors use cone opsins while rod photoreceptors use rhodopsin. Photons of light cause 11-cis-retinal isomerization into all-trans-retinal, changing the conformation of the opsin GPCR and starting the signal transduction cascade. The isomerization of retinoid uses oxidation and reoxidation reactions, therefore, photopigments are particularly susceptible to oxidative damage. Preventing the accumulation of damage in photoreceptor outer segments becomes critical. As a result, outer segment disks and photopigments are constantly replenished and renewed, with an estimated 70 rhodopsin molecules being synthesized and delivered each second[5]. The addition of newly synthesized photoreceptor proteins occurs at the basal side of the outer segments through vesicular fusion. As a result, older and damaged outer segment components accumulate at the most apical end of the outer segments where they get phagocytosed by macrophages found in the retinal pigment epithelium.

The relay of visual information is initiated by photoreceptors through glutamate release at specialized "ribbon" synapses. A proteinaceous ribbon structure is specialized to the synapses of sensory neurons such as cochlear hair cells, vestibular hair cells, photoreceptors, and retinal bipolar cells[6]. This synaptic ribbon is tethered near the active zone and allows docking of vesicles in an arrayed formation to increase vesicle pool sizes[7]. Uniquely, photoreceptors are in the depolarized state at default and constantly releasing glutamate. Decreases to glutamate release occurs after the receipt of a light signal and as a result, synaptic vesicle exocytosis at photoreceptor ribbon synapses occurs in massive quantities at the magnitudes of thousands of vesicles per second[7].

Due to vesicle trafficking processes occurring at multiple regions of photoreceptors, vesicular transport/exocytosis is considered to play two critical roles in retinal photoreceptor cells: (1) delivery of outer segments components to renew old and damaged components, (2) glutamate neurotransmitter release for synaptic transmission[3,8,9]. However, the underlying vesicular transport and exocytosis mechanisms that mediate both processes in photoreceptors remain largely unclear. Elucidating these processes will be important for addressing diseases with impairments to outer segment protein trafficking and/or synaptopathies stemming from defective photoreceptor synaptic transmission.

It is well established that synaptic vesicle exocytosis in "conventional" central nervous system neurons is dependent on a three-protein exocytosis complex consisting of a vesicular v-SNARE, synaptobrevin-2, and two target t-SNAREs, syntaxin-1 and SNAP-25[10,11]. However, unlike conventional neurons, photoreceptors do not express syntaxin-1. Notably, syntaxin-3, which is a ubiquitously found syntaxin isoform, being utilized by the specialized photoreceptor neurons in the eye is surprising[12,13].

Despite the importance of exocytosis in photoreceptors, the identity and function of the SNAP family member which is the other obligate t-SNARE has yet to be conclusively defined.

SNAP-25 mRNA expression has not been detected in adult photoreceptors, although it is expressed in other second-order retinal neurons and ganglion cells[14]. However, at the protein level, SNAP-25 presence in photoreceptor have been supported[15–18]. Although, immunofluorescence experiments employing SNAP-25 antibodies revealed distinct localization patterns based on the specific antibody utilized[19]. On the other hand, ubiquitously expressed SNAP-23 has also been shown to be present in the retina with enrichment within the presynaptic ribbon fraction[20]. SNAP-23 is important for exocytosis in immune cells[21–24] and pancreatic beta cells;[25,26] in pancreatic beta cells SNAP-23 plays a negative role[26,27], and it is unclear whether these modulatory roles or other functions are carried out by SNAP-23 in the retina. SNAP-23 preferentially binds with syntaxin-3 within immune cells[25]. Although it has been shown before that SNAP-25 does bind with syntaxin-3, it is with less affinity than syntaxin-1A[28,29]. Additionally, photoreceptor cells utilize syntaxin-3B which is a splice isoform of syntaxin-3 and syntaxin-3B binds SNAP-25[13,30–34]. As a result, whether photoreceptor neurons employ the ubiquitously expressed SNAP-23 or neuronal SNAP-25 within specialized photoreceptor neurons and consequently, the functional importance of SNAP-23 and SNAP-25 in photoreceptors has not been unequivocally demonstrated in vivo.

In the present study, we investigated the requirement for SNAP-23 and SNAP-25 by generating photoreceptor-specific knockouts of SNAP-23 and SNAP-25. We show that SNAP-23 is dispensable for photoreceptor function whereas SNAP-25 is essential for photoreceptor development and maintenance. Using a combination of methods ranging from in situ hybridization to antibody staining, we show that SNAP-25 expression is developmentally regulated, initially localized to photoreceptor nuclei at P6, and trafficked to the inner segment starting at P9 when outer segments begin developing. The timing of this transport coincided with the onset of the photoreceptor degeneration in SNAP-25 conditional knockout mice, where the photoreceptor nuclear layer is almost completely absent by P14. Together, we show the importance of SNAP-25 in photoreceptors.

## Results

**Conditional SNAP-23 inactivation in photoreceptors with CRX-Cre does not impair visual function.** We used in situ hybridization and IHC to complementarily investigate SNAP-23 mRNA expression and protein expression to confirm SNAP-23 presence in photoreceptors. Due to processes such as post-transcriptional modifications to mRNA and protein processing such as ubiquitination processes, mRNA expression level does not always correlate with protein expression[35]. We detected SNAP-23 mRNA in the inner nuclear layer (INL) and ganglion cell layer (GCL) and weaker SNAP-23 expression in the outer nuclear layer (ONL) where photoreceptors are localized (Fig. 1a). SNAP-23 mRNA signal was only observed by our antisense probe, but not by the sense probe. IHC showed that SNAP-23 was localized to postsynaptic cells in the outer plexiform layer staining in the outer plexiform layer (OPL), as the signal did not overlap with staining for PSD95 (Fig. 1b), which marks photoreceptor synaptic terminals.

To examine the role of SNAP-23 in photoreceptors, we generated SNAP-23 conditional knockout (cKO) mice by crossing SNAP-23 flox mice[36,37] with CRX-cre mice[38]. CRX-cre is active starting at E12.5 and is selectively expressed in photoreceptors

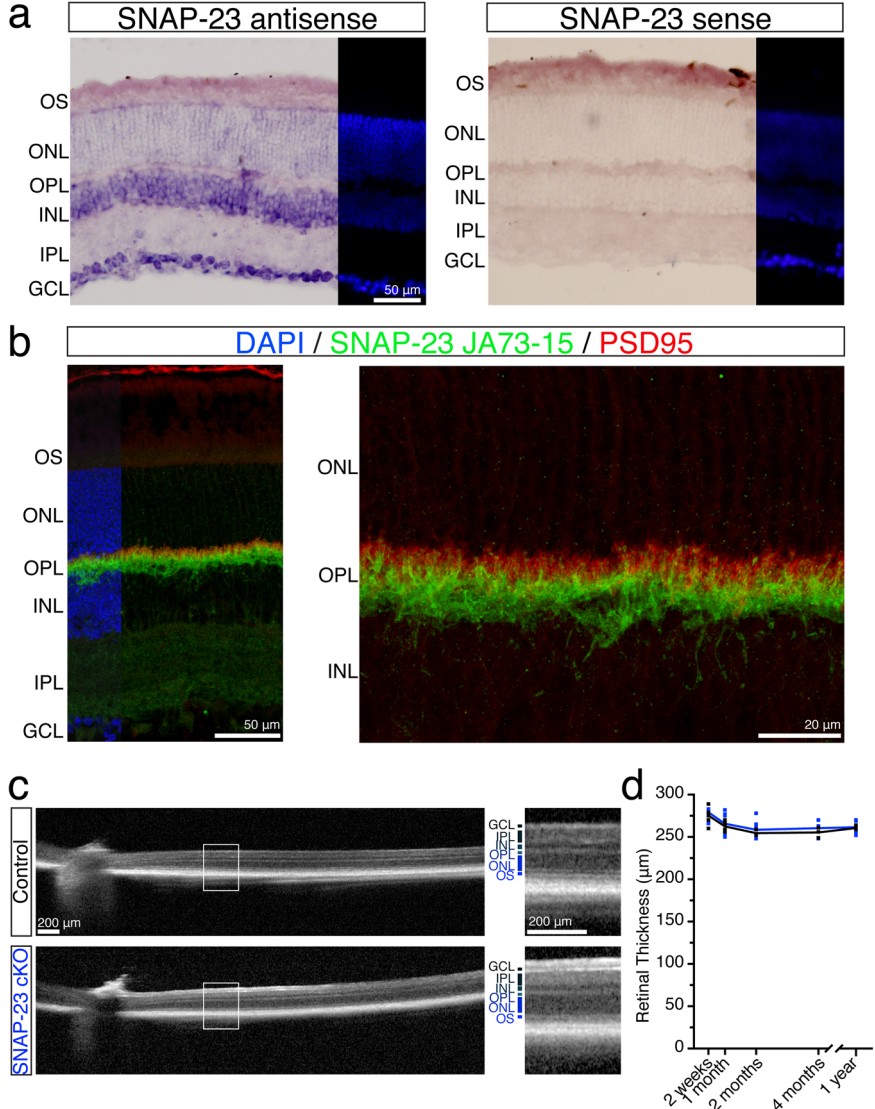

**Fig. 1 Removal of SNAP-23 from photoreceptors using CRX-cre does not affect retinal thickness. a** In situ hybridization using antisense probe of SNAP-23 exons 2–8 identifies SNAP-23 presence strongly in the inner nuclear layer (INL) and ganglion cell layer (GCL), SNAP-23 signal is weakly observed in the outer nuclear layer (ONL) where photoreceptors are. No signal was observed by the SNAP-23 sense probe. Scale bar = 50 μm. **b** Immunostain of SNAP-23 using rabbit monoclonal antibody (green) finds SNAP-23 strongly present in the post-synaptic region of the outer plexiform layer (OPL) and no colocalization with presynaptic PSD95 (red). Scale bar = 50 μm (left) and 20 μm (higher magnification, right). **c** Optical coherence tomographs of SNAP-23 control and SNAP-23 cKO retinas. No change to retina thickness was observed. Scale bar = 200 μm. **d** Tracking of retinal thickness 1.0–1.5 mm away from the optic nerve over the course of 1 year for control and SNAP-23 cKO mice. No change to retina thickness was observed over time (minimum $n = 3$ for all time points for both control and SNAP-23 cKO mice).

and a subset of bipolar cells[13,38]. Retinal thickness in SNAP-23 cKO and Cre- control littermates was assessed by optical coherence tomography (OCT), however, no changes were observed in mice up to one year of age (Fig. 1c, d). Rod and cone photoreceptors in SNAP-23 cKO mice appeared morphologically normal (Supplementary Fig. 1A). Photoreceptor synaptic morphology also appeared normal as staining for the synaptic ribbon (marked by RIBEYE) demonstrated proper stratification above post-synaptic rod bipolar cells (marked by PKCα) in both control and SNAP-23 cKO mice (Supplementary Fig. 1B).

Using electroretinograms (ERGs), we tested the ability of SNAP-23 deficient photoreceptor cells to respond to light. Dark-adapted electroretinograms were first performed to measure rod photoreceptor responses. We find that with increasing light intensities, the amplitude of the b-wave increased in both control and SNAP-23 cKO mice (Fig. 2a, b). With increasing light

stimulation, the synchronicity of photoreceptors and the latency of the b-wave decreased in both control and SNAP-23 cKO mice (Fig. 2c). Following a 10-min light-adaptation period, photopic ERGs were recorded to determine cone responses (Fig. 2d). Similar to the scotopic ERGs, increasing light intensity increased the b-wave amplitude while decreasing the b-wave latency (Fig. 2e, f).

Finally, we tested the visual acuity of control and SNAP-23 cKO mice using optokinetic tracking (OKT). The OKT is a well-defined task wherein mice are placed in a chamber with four monitors presenting rotating black and white vertical sine-wave gratings of varying thicknesses[39]. Mice reflexively follow the rotation of the bars until the bars becomes too fine to be distinguishable to them. Control and SNAP-23 cKO mice were able to track to a similar spatial frequency, demonstrating no deficits in visual acuity in accordance with the unimpaired ERGs.

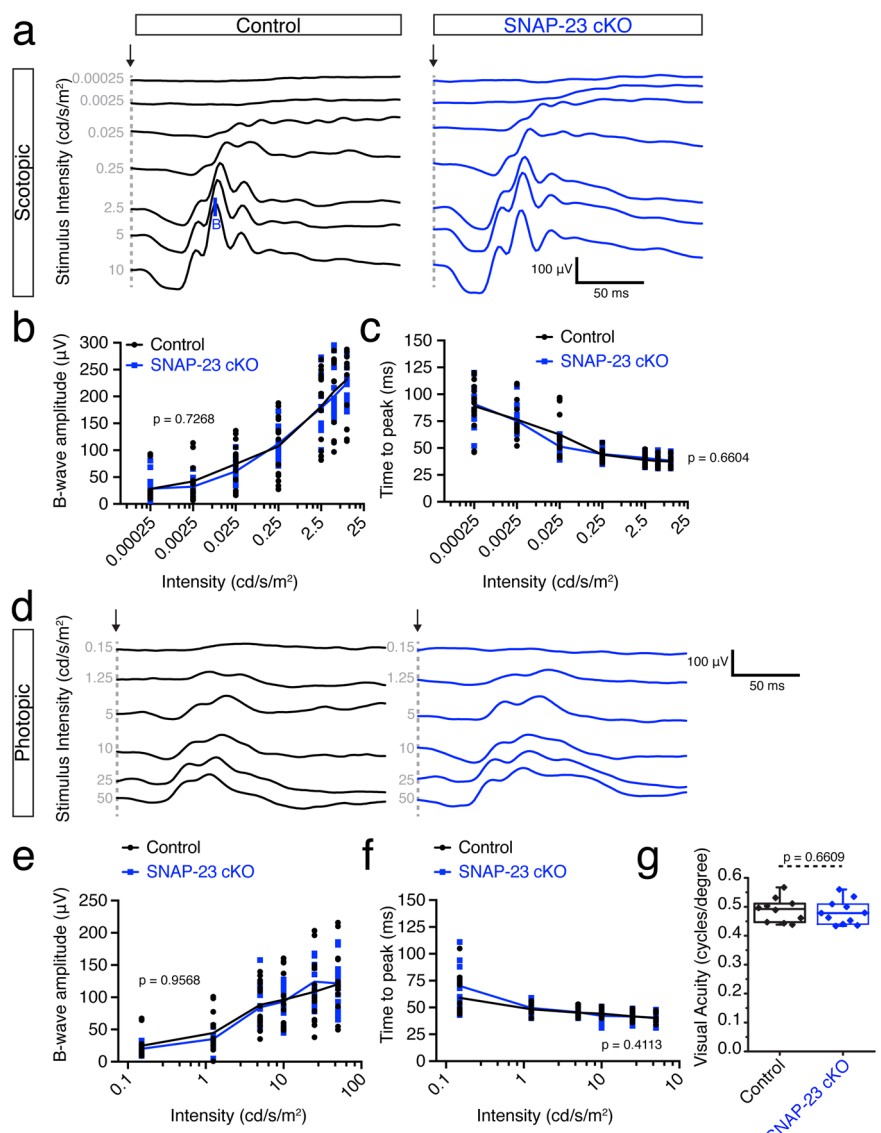

**Fig. 2 SNAP-23 conditional knockout does not impair visual function. a** Representative scotopic electroretinogram traces across seven indicated light intensities. Arrow indicate stimulation onset, B indicates b-wave. Increase to b-wave amplitude increase and decrease to b-wave onset was observed for both control and SNAP-23 cKO mice. Calibration bars represent 50 ms (horizontal), and 100 μV (vertical). **b** Quantifications of scotopic b-wave amplitudes ($n = 7$ mice for both control and SNAP-23 cKO for all quantifications; two-way ANOVA; $F_{(1, 26)} = 0.1248$; $p = 0.7268$). **c** Quantifications of scotopic time to peak of ERG b-wave. Two-way ANOVA; $F_{(1, 26)} = 0.1976$; $p = 0.6604$. **d** Representative photopic electroretinograms across 6 light intensities. Arrow indicates stimulation onset. Calibration bars represent 50 ms (horizontal), and 100 μV (vertical). **e** Quantification of photopic b-wave amplitudes; two-way ANOVA; $F_{(1, 26)} = 0.003$; $p = 0.9568$. **f** Quantifications of photopic time to peak of ERG b-wave; two-way ANOVA; $F_{(1, 26)} = 0.6973$; $p = 0.4113$. **g** Visual acuity assessed by measuring optokinetic tracking response using CerebralMechanics OptoMotry system. No change to visual acuity is observed in SNAP-23 cKO mice compared to control following SNAP-23 removal from photoreceptors. ($n = 10$ for control, $n = 11$ for SNAP-23 cKO; two-sample two-sided $t$ test; $t_{(19)} = 0.4456$; $p = 0.6609$). Box-and-whisker plot represents the median (central line), 25th–75th percentile (bounds of the box), and min/max (whiskers).

Overall, weakly expressed *SNAP-23* in photoreceptor cells seems to be functionally dispensable.

**The expression and transport of *SNAP-25* mRNA is developmentally regulated.** Our results found that ubiquitously expressed SNAP-23 is unlikely to be the essential photoreceptor SNAP isoform, therefore we considered the neuronally expressed SNAP-25 as the potential photoreceptor SNAP-25 isoform and revisited the expression of *SNAP-25* mRNA and protein within photoreceptors. Previous mRNA in situ studies failed to find *SNAP-25* mRNA in adult photoreceptors while RNA-sequencing

data analysis found that *SNAP-25* was expressed in developing and adult photoreceptors[14,40]. First, we looked at *SNAP-25* mRNA expression during development on postnatal days P4, P6, P9, P13, and P17. We find that *SNAP-25* mRNA expression was very abundant in the outer nuclear layer at P6 and P9 (Fig. 3a). Following the development of photoreceptor outer segments at P9, we also note that the location of the *SNAP-25* mRNA changed, and *SNAP-25* mRNA was transported to the inner segments of the photoreceptors, where translation occurs[41,42]. At adult ages, we were able to observe *SNAP-25* mRNA signal in the inner segments of photoreceptors (Fig. 3b), and consistent with this, RNAscope found a large abundance of *SNAP-25* transcript in the

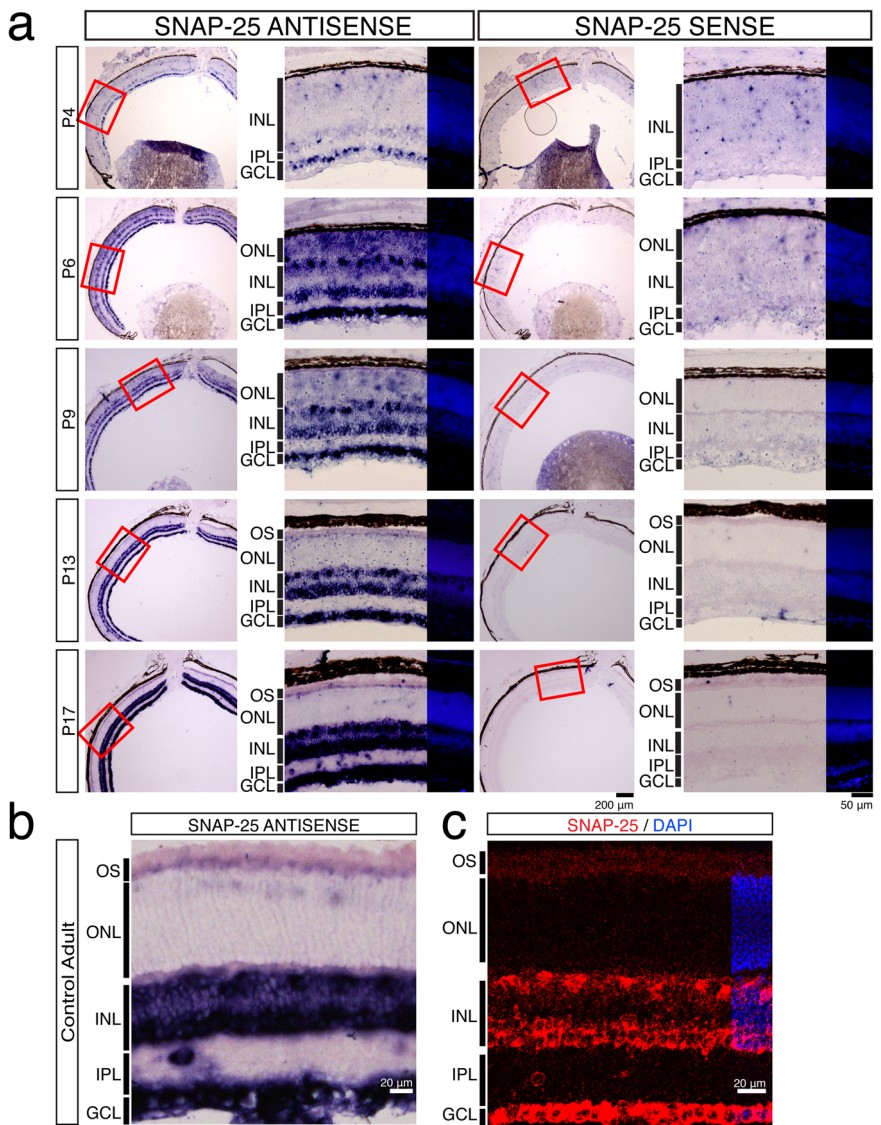

**Fig. 3 SNAP-25 mRNA is developmentally regulated and transported to the inner segments. a** In situ hybridization using *SNAP-25* antisense and sense probes at postnatal day 4 (P4), P6, P9, P13, and P17. Photoreceptor *SNAP-25* mRNA is developmentally regulated and expression is observed throughout the outer nuclear layer (ONL) starting at P6. At P9, *SNAP-25* transcript begins to be trafficked to the inner segments of the photoreceptors. By P17, *SNAP-25* transcript can no longer be observed in photoreceptor nuclei and is in the outer segments. Inner nuclear horizontal, bipolar, amacrine, and ganglion cells express high amounts of *SNAP-25* mRNA at all ages. Scale bar = 200 μm (whole sections), and scale bar = 50 μm (magnified region). **b** Adult in situ hybridization using the same *SNAP-25* antisense probe as (**a**). In adult mice, SNAP-25 transcript can still be found in the inner segments of the photoreceptor. Scale bar = 20 μm. **c** RNAscope for SNAP-25 also identifies *SNAP-25* expression in the photoreceptor inner segments, horizontal cells, amacrine cells, and ganglion cells in adult retina sections. Scale bar = 20 μm.

inner nuclear layer and ganglion cell layer (Fig. 3c). Therefore, we suspect the *SNAP-25* signal observed in the inner segments was previously missed due to the massive amount of *SNAP-25* present in the inner nuclear layers and the ganglion cell layer and reconcile the lack of *SNAP-25* detected by in situ hybridization previously with the presence of *SNAP-25* mRNA reported with RNA-seq data.

Next, we examined the SNAP-25 protein expression using a well-defined anti-SNAP-25 antibody. Monoclonal SNAP-25 antibody SMI81 identified SNAP-25 colocalization with presynaptic PSD95 (Supplementary Fig. 2A). We could also observe SNAP-25 signal surrounding photoreceptor cell nuclei and in the photoreceptor outer segments, indicating SNAP-25 is present in photoreceptors. We also tested an additional SNAP-25 antibody (EP3274, Supplementary Fig. 2B), and again we were able to observe SNAP-25 signal around photoreceptor cell bodies and

outer segments. Therefore, we find that SNAP-25 protein is indeed found in photoreceptors and that *SNAP-25* mRNA is located within photoreceptor inner segments during adulthood.

**Removal of SNAP-25 from photoreceptors leads to photoreceptor degeneration and abolishes visual function.** We again used CRX driven cre-recombinase to conditionally remove SNAP-25 from photoreceptors by crossing with SNAP-25 flox mice. Similar to SNAP-23 cKO, we performed optical coherence tomography (OCT) experiments. The removal of SNAP-25 dramatically thinned the retinas of the SNAP-25 cKO mice (Fig. 4a, b). Retinas were thin at 2 weeks when eye-opening occurred and inner retinal cells (e.g., bipolar cells, horizontal cell, amacrine cells, ganglion cells, etc.) continued to further degenerate over the course of 4 months. ERGs were next performed, and while the b-wave appeared around the third scotopic stimulation intensity for

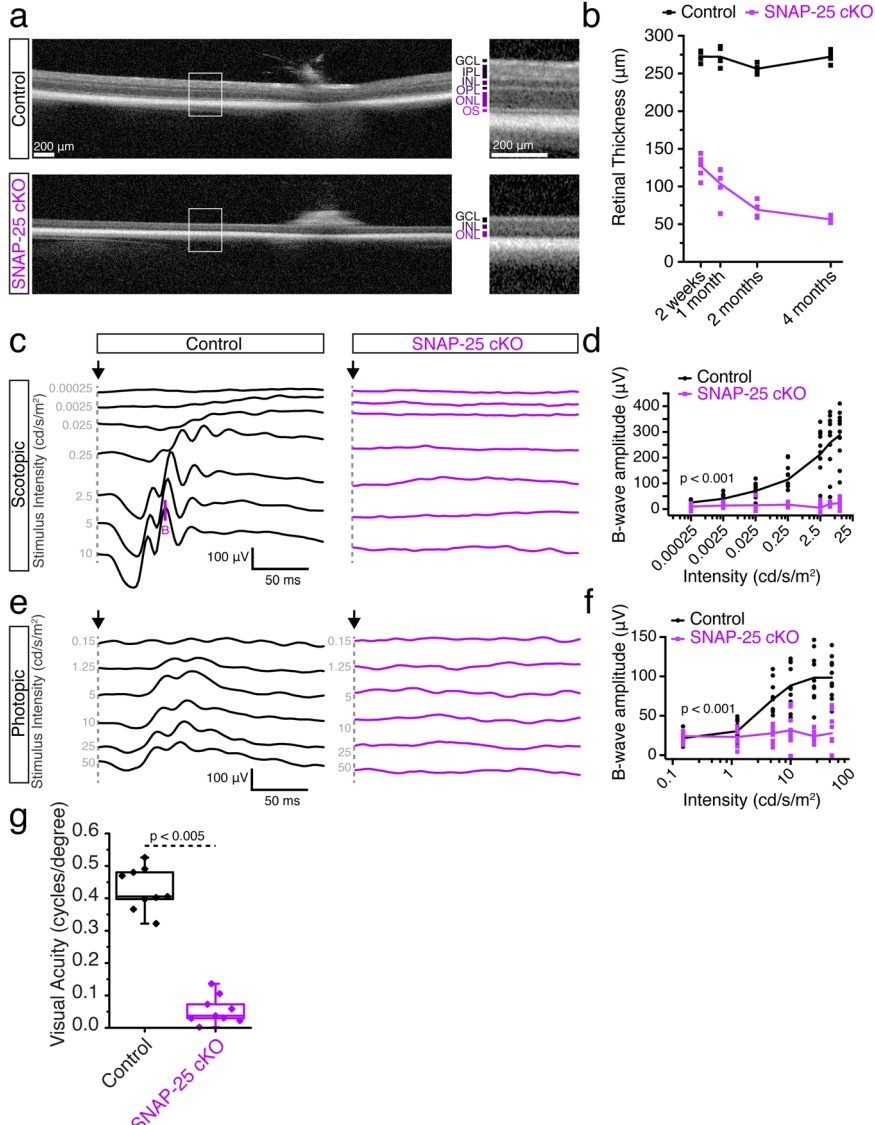

**Fig. 4 Removal of SNAP-25 from photoreceptors using CRX-cre severely decreases retinal thickness and abolishes visual functioning. a** Optical coherence tomographs of SNAP-25 control and SNAP-25 cKO retinas. A severe decrease to retinal thickness is observed in SNAP-25 cKO retinas. Scale bar = 200 μm. **b** Tracking of retinal thickness over the course of 4 months for control and SNAP-25 cKO mice. SNAP-25 retinas are thin at 14 days and continue to further degenerate over time. (minimum n = 4 for all time points). **c** Representative scotopic electroretinogram traces across 7 light intensities. Arrow indicate stimulation onset, B indicates b-wave. SNAP-25 cKO mice exhibit no responses to light stimulation at any light intensity at postnatal day 30. Calibration bars represent 50 ms (horizontal), and 100 μV (vertical). **d** Quantifications of scotopic B-wave amplitudes. (n = 5 mice for both control and SNAP-25 cKO for all quantifications; two-way ANOVA; $F_{(1, 18)} = 47.49$; p < 0.0001. **e** Representative photopic electroretinograms across 6 light intensities. Arrow indicates stimulation onset. Calibration bars represent 50 ms (horizontal), and 100 μV (vertical). **f** Quantification of photopic B-wave amplitudes; two-way ANOVA; $F_{(1, 18)} = 36.61$; p < 0.0001. **g** Visual acuity assessed by measuring optokinetic tracking response using CerebralMechanics OptoMotry system. Severe decrease to visual acuity is observed in SNAP-25 cKO mice compared to control following SNAP-25 removal from photoreceptors. (n = 9 for both and SNAP-25 cKO; two-sample two-sided t test: $t_{(16)} = 14.20$; p < 0.005). Box-and-whisker plot represents the median (central line), 25th–75th percentile (bounds of the box), and min/max (whiskers).

control mice (Fig. 4c, d) no apparent b-wave was observed at any stimulation intensity in SNAP-25 cKO mice. Similarly, no b-wave appeared for photopic ERGS in SNAP-25 cKO mice (Fig. 4e, f). ERGs were repeated on mice shortly after eye opening at P16 when photoreceptor synapses are established[43]. Scotopic and photopic ERGs recorded at P16 found that light stimulation failed to elicit both a-wave or b-wave responses in SNAP-25 cKO (Supplementary Fig. 3). SNAP-25 cKO also mice showed no response in the OKT at any spatial frequency (Fig. 4g), demonstrating no visual acuity in accordance with the complete abolishment of light response in SNAP-25 cKO mice.

Since we observed changes to OCT and no electrophysiological or behavioral responses by ERG or OKT by P16, we propose a role for SNAP-25 photoreceptor development and maintenance. We began our investigation at P9 when we first noted *SNAP-25* mRNA being trafficked to the photoreceptor inner segments. First, we repeated SNAP-25 antibody staining using both SNAP-25 antibodies on P9 control and SNAP-25 cKO retinal sections. Similar to older ages, we saw SNAP-25 protein expression throughout the control retina (Fig. 5a, b), in photoreceptors, second-order cells, and ganglion cells. In strong contrast, the photoreceptor SNAP-25 signal was completely absent in

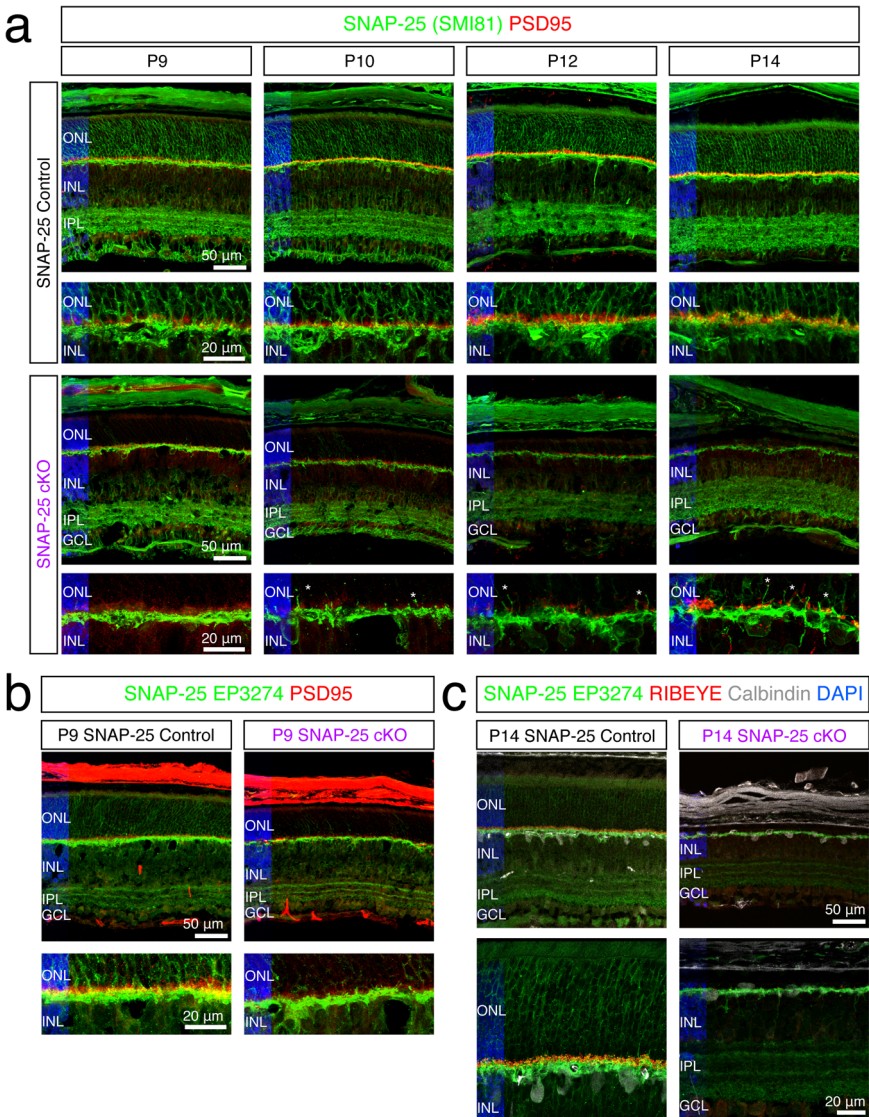

**Fig. 5 Efficient and specific removal of SNAP-25 from photoreceptors using CRX-cre. a** Antibody staining for SNAP-25 (green) using mouse monoclonal SMI-81 antibody in control retina finds SNAP-25 expressed in the sclera, the outer nuclear layer, outer plexiform layer, inner nuclear layer, inner plexiform layer, and ganglion cell layers. Retina region for all represented images is 0.5–1 mm away from optic nerve. SNAP-25 is located surrounding photoreceptor nuclei, in photoreceptor outer segments, and also photoreceptor synaptic terminals (PSD95, red). In SNAP-25 cKO mice, SNAP-25 is eliminated from photoreceptors, but inner retinal SNAP-25 remains intact. Thinning of the photoreceptor outer nuclear layer can be observed in the SNAP-25 cKO mice and ectopic sprouting of the postsynapse can be observed as early as P10 in SNAP-25 cKO mice (white asterisk). Scale bar = 50 μm and scale bar = 20 μm (higher magnification lower panel). **b** Antibody staining for SNAP-25 (green) using rabbit monoclonal EP3274 antibody in control and SNAP-25 cKO retina. SNAP-25 surrounds photoreceptor nuclei, is present in photoreceptor outer segments, and colocalized with photoreceptor synaptic terminals (PSD95, red). SNAP-25 protein is strongly decreased in SNAP-25 cKO photoreceptors. The sclera is also strongly stained by the PSD95 antibody. Scale bar = 50 μm and scale bar = 20 μm (higher magnification lower panel). **c** Antibody staining for SNAP-25 (green), synaptic ribbons (RIBEYE, red), horizontal cells (white) at P14. Despite SNAP-25 removal from photoreceptors and photoreceptors being completely absent, postsynaptic SNAP-25 is unaffected and remains in horizontal cells and the inner plexiform layer confirming specificity of CRX-cre removal. Scale bar = 50 μm and scale bar = 20 μm (higher magnification lower panel).

SNAP-25 cKO retina and no SNAP-25 signal was observed in the outer segments, surrounding photoreceptor nuclei, or at synaptic terminals by both antibodies (Fig. 5a, b). When we tracked retina thickness up until the day of eye-opening, we found severe degeneration where photoreceptors are almost completely gone by P14 (Fig. 5a). The loss of photoreceptors also resulted in the formation of synaptic sprouts of the postsynapse as early as P9 and loss of presynaptic PSD95 (Fig. 5a) and RIBEYE signal (Fig. 5c). However, postsynaptic SNAP-25 present in second-order cells and ganglion cells was unaffected by our photoreceptor-specific SNAP-25 cKO (Fig. 5a, b), and at P14,

horizontal cells continued to abundantly express SNAP-25 despite most photoreceptors already being gone (Fig. 5c).

We picked P9 mice to further observe retinal morphology by transmission electron micrographs as SNAP-25 deletion from photoreceptors is clearly observed by this age (Fig. 5a), but prior to the onset of degeneration (Fig. 6a). Mouse photoreceptors develop with a conventional nuclear architecture with multiple regions of dense heterochromatin[44,45] which invert with maturity as characteristic of nocturnal species. At P9, SNAP-25 control photoreceptors showed the expected conventional nuclear architecture (Fig. 6a). However, P9 SNAP-25 cKO photoreceptors

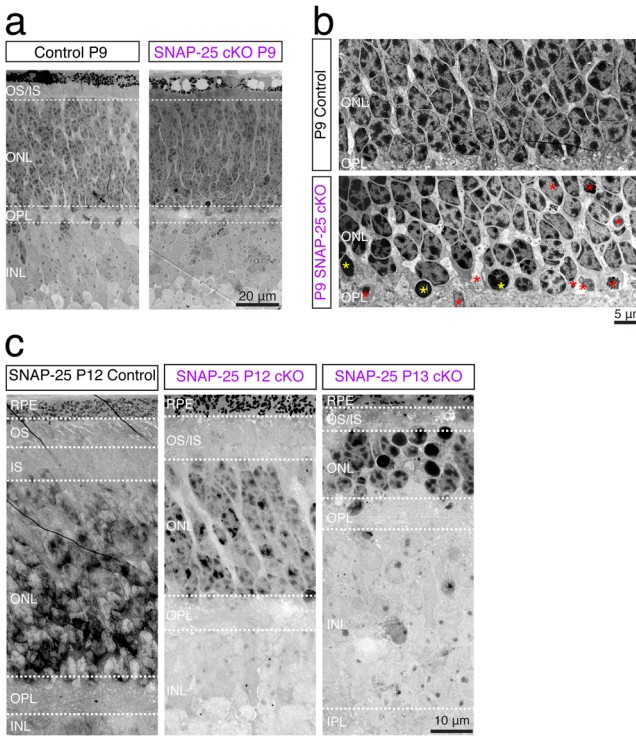

**Fig. 6 Nuclear changes associated with the degeneration of photoreceptors. a** Retina thickness looks comparable between control and SNAP-25 cKO groups at postnatal day 9 (P9). n = 2 SNAP-25 P9 cKO mice were examined by EM. Scale bar = 20 μm. **b** Electron micrographs of control and SNAP-25 cKO retinas at postnatal day 9 (P9). At P9, photoreceptor nuclei exhibit conventional chromatin organization and have peripheral heterochromatin clusters. Chromatin structure is altered in unhealthy photoreceptors and uniformly dark condensed nuclei (yellow asterisks) can be observed in SNAP-25 cKO retinas. Other abnormal photoreceptors can be observed (red asterisks) and have features such as vacuolation, loss of membrane integrity, and fragmentation. Features were observed in both SNAP-25 P9 cKO mice examined by EM. Scale bar = 5 μm. **c** Electron micrographs of control and SNAP-25 cKO retinas at P12 and SNAP-25 cKO retinas at P13. n = 2 SNAP-25 P12 and P13 cKO mice were examined by EM. Scale bar = 10 μm.

exhibited abnormal nuclear architecture based on nuclei with condensation of the chromatin (Fig. 6b, yellow asterisks), as well as nuclei with fragmentation and vacuolation along the basal outer nuclear layer (Fig. 6b, red asterisks). We corroborated the pattern of death by immunohistology (Supplementary Fig. 4, red arrows), and observed pyknotic nuclei and changes in nuclear permeability that allowed greater amount of DAPI binding to photoreceptor nuclei undergoing cell death[46,47]. In contrast to the almost unchanged outer nuclear layer thickness that is observed in P9 in SNAP-25 cKO mice (Fig. 6a), severe degeneration was observed in SNAP-25 cKO photoreceptors at P12 and P13 by electron micrographs (Fig. 6c).

**Absence of SNAP-25 in photoreceptors impairs outer segment trafficking in addition to outer segment and synaptic morphology.** At P9, we could observe the early formation of photoreceptor outer segments, and in control mice, the inner segment region has discrete individual bundles (Fig. 7a and Supplementary Fig. 5). However, in SNAP-25 cKO mice, while these discrete formations were present, the presence of vacuolated inner segments were observed and resulted in the inner segment region being more disorganized (Fig. 7a and Supplementary Fig. 5).

At P9, although photoreceptor synapses are not fully developed[12], we were able to observe defined photoreceptor synaptic terminals with synaptic elements (Fig. 7b, c). SNAP-25 cKO mice were able to develop this synaptic layer, however, the boundary was much more chaotic (Fig. 7b). Additionally, photoreceptor terminals exhibited much vacuolation to them and many photoreceptor nuclei next to the terminal showed irregular chromatin structure (Fig. 7b). The postsynaptic region also showed distortions at P9 in SNAP-25 cKO animals (Fig. 7b, c), which we attribute to the sprouting of dendrites normally postsynaptic to photoreceptors, and was consistent with previous immunohistology observations (Fig. 5a, asterisks). At P9, control photoreceptor synapses contained synaptic ribbons, some of which were beginning to form proper postsynaptic connections (Fig. 7c, green arrows). While this was also observed in some SNAP-25 cKO photoreceptor terminals, abnormal ribbons that were duplicated were also present (Fig. 7c, yellow arrows). Spherical ribbons, which are a feature of developing ribbon synapses[48,49], were observed in both P9 control and SNAP-25 cKO animals. However, P9 SNAP-25 cKO animals had an increase in the proportion of spherical and immature ribbons (Supplementary Fig. 6). Vesicle size was enlarged in SNAP-25 cKO photoreceptor terminals, though the vesicles were properly associated with their ribbons in both SNAP-25 control and SNAP-25 cKO mice and consistent with the properly localized synaptobrevin-2 immunostaining we observed (Supplementary Fig. 7).

The extreme time course of degeneration wherein the retina goes from completely normal at P9 to having very few photoreceptors by P14 corresponds with when outer segment development initiates (P8) and photoreceptors begin to form synapses (P7 cones, P9 rods) and gives us a unique opportunity to investigate the order of development and degeneration[12,50]. While we saw disruptions to both outer segments and synaptic terminals in electron micrographs, we could not gauge which process was affected first. Thus, we proceeded with immunostains for photoreceptor pigments. Photopigments are synthesized at the soma of photoreceptors and transported to the outer segment by vesicular trafficking during development and in control mice at P9, properly trafficked rhodopsin was found abundantly in the apical region of the photoreceptors (Fig. 8a). At higher magnifications, there was clearly a gap between the outer limiting membrane (dashed line) and where rhodopsin was observed, suggesting that rhodopsin was properly trafficked out of the inner segments into the outer segments. In P9 SNAP-25 cKO mice, the majority of rhodopsin was still located in the apical region of photoreceptors (Fig. 8a), however, rhodopsin signal was also observed throughout the outer nuclear layer and closer to the synaptic region of photoreceptors suggesting improper trafficking. SNAP-25 cKO photoreceptors also had rhodopsin in regions of the inner segment adjacent to the outer limiting membrane, further demonstrating mistrafficking. We also note that at this age, SNAP-25 cKO mice had some cones where cone pedicles were retracting into the ONL (Fig. 8a). We further stained for short wavelength cone opsin using S-opsin in P9 control and SNAP-25 cKO mice (Fig. 8b). Control SNAP-25 mice had abundant S-opsin located in the apical region of photoreceptors, on the other hand, SNAP-25 cKO mice had reduced number of S-opsin (Fig. 8b, c) and there were also S-opsin stains within the outer nuclear layer throughout the whole cone photoreceptor (Fig. 8b). Remaining apical S-opsin were also shorter and wider, suggesting photoreceptors were swollen or did not have their usual thin ciliary morphology. At the same time, we also stained for the synaptic ribbon using RIBEYE (Fig. 8b). At P9, RIBEYE was observed along the outer plexiform above rod bipolar cells in both control and SNAP-25 cKO retinas. While there was mislocalized rhodopsin and S-opsin observed in our SNAP-25

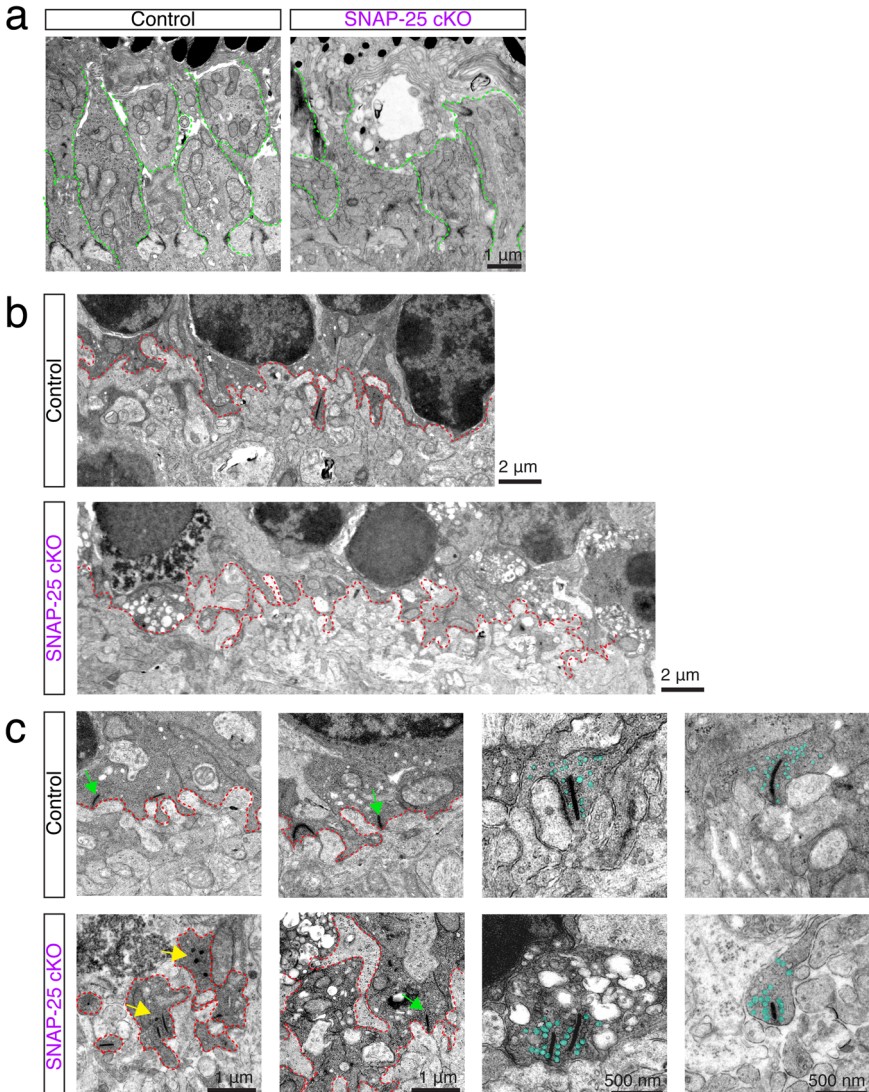

**Fig. 7 Outer segment and synaptic degenerations observed in P9 SNAP-25 cKO retina. a** Representative images of photoreceptor outer segments at postnatal day 9 in control and SNAP-25 cKO retinas. Control retinas have discrete beginnings of inner segments (outlined in green), which were also evident in SNAP-25 cKO retinas. However, these bundles were swollen and vacuolated in the SNAP-25 cKO retinas. Representative pictures selected from $n = 2$ mice. Scale bar = 1 μm. **b** Photoreceptor terminals and synaptic boundary (red dashed line) in control and SNAP-25 cKO retinas at P9. The synaptic boundary could be clearly identified in both samples, however in the SNAP-25 cKO samples, the boundary was more chaotic and had signs of vacuolation and abnormal photoreceptor nuclei were present nearby. Scale bar = 2 μm. **c** Close-ups of the synaptic region and synapses in control and SNAP-25 cKO mice. Normal synaptic ribbons were observed in both control and SNAP-25 cKO samples (green); however, spherical ribbons (yellow) and vacuolation were also observed in the SNAP-25 cKO sample. Scale bar = 1 μm. Synaptic vesicles (teal) appeared properly associated with synaptic ribbons. Scale bar = 500 nm.

cKO photoreceptors, there were still photopigments observed in the outer segments as expected. We conclude that exocytosis plays roles in both photopigment trafficking in the outer segments and synaptic vesicle exocytosis.

## Discussion

Ribbon synapses are specialized synaptic structures that are present at the terminals of photoreceptors, bipolar cells, vestibular hair cells, and cochlear hair cells. Unexpectedly, one study showed that exocytosis in mouse inner hair cells (IHC) is insensitive to neuronal SNARE-cleaving neurotoxins and SNAP-25 knockout had little effect on exocytosis, suggesting that IHC exocytosis apparently occurred independently of neuronal SNAREs and makes use of other SNAREs[51]. Combined with

findings that syntaxin-3 is involved with exocytosis at photoreceptor synapses[13], we originally expected a combination of non-neuronal t-SNAREs to participate in photoreceptor development and survival. However, we find that removal of SNAP-23 had no consequences to photoreceptor morphology or function (Figs. 1 and 2). A more recent study showed that IHC-specific knockout of SNAP-25 caused severe reductions in inner hair cell transmitter release and synaptic degeneration[52], in contrast to the previous report[51]. Our results of SNAP-25 removal from photoreceptor cells are more consistent with the more recent study, perhaps due to the early postnatal removal of SNAP-25 utilized in both studies. Thus, although SNAP-25 binds with syntaxin-3 at a lower affinity compared to syntaxin-1[30,32], the neuronal SNARE SNAP-25 is still an obligate t-SNARE supporting photoreceptor functioning.

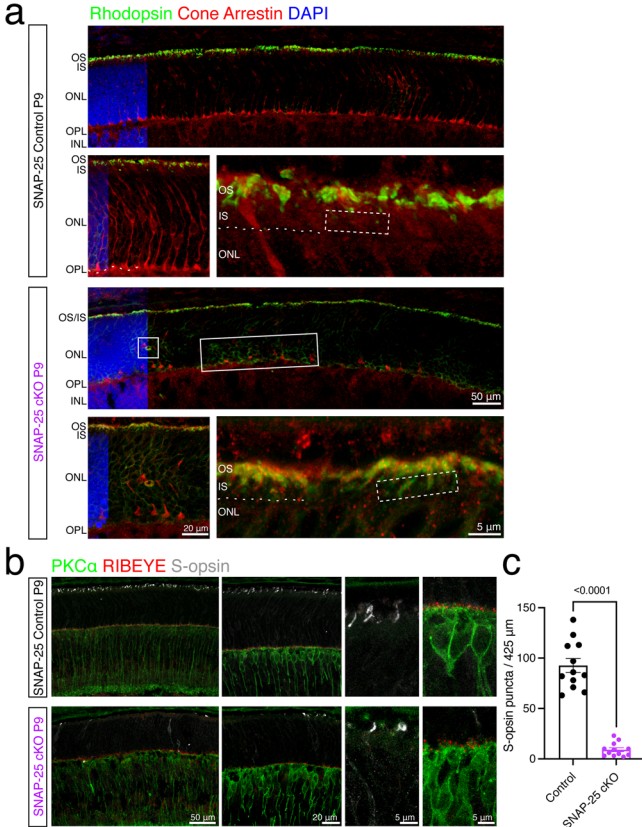

**Fig. 8 Photopigment trafficking impairments in SNAP-25 cKO mice.**
**a** Antibody staining for rhodopsin (green) and cones (red) in P9 control and SNAP-25 cKO retina. Rhodopsin is observed in the apical photoreceptor region and at higher magnifications can be found located with a gap above the outer limiting membrane (OLM, white dashed line) exhibiting proper trafficking out of the photoreceptor inner segments. Rhodopsin trafficking was altered in SNAP-25 cKO photoreceptors and rhodopsin was present within the inner segments (white dashed box) and in the outer nuclear layer (white box). Also noted is that cone pedicles did not properly stratify to the outer plexiform like in control retina or were retracting into the outer nuclear layer. Similar phenotype was observed across $n = 3$ mice per group. Scale bar = 50 μm (low magnification), and scale bar = 20 μm and 5 μm (higher magnifications). **b** Antibody staining against rod bipolar cells (PKCα, green), synaptic ribbons (RIBEYE, red), and S-opsin (white) finds that at P9, while synaptic ribbons are plentiful, cone opsins are very sparse. Retina region for represented images is ventral retina, approximately 0.5 mm away from optic nerve. Similar phenotype was observed across $n = 3$ mice per group. Scale bar = 50 μm (low magnification), and scale bar = 20 μm and 5 μm (higher magnifications). **c** Quantifications of S-opsin puncta in P9 control SNAP-25 cKO mice. Total number of puncta per 425 μm of retina were counted. Two-sample two-sided $t$ test: $t_{(21)} = 11.63$; $p < 0.001$. $n = 3$ control and $n = 3$ SNAP-25 cKO mice aged within 24 h of postnatal day 9 were used. Four images from different retinal sections were quantified from each mouse. Error bars represent SEM.

Here, we provide clear evidence that SNAP-25 is critical during photoreceptor development for the delivery of outer segment proteins and maintaining the ribbon synapse. We find that *SNAP-25* mRNA is developmentally regulated and gets trafficked to photoreceptor inner segments starting around postnatal day 9 (Fig. 3). Removal of SNAP-25 from photoreceptors causes the death of the photoreceptor layers and loss of light response and vision (Fig. 4). Photoreceptors developed normally up to P9 and then died off in the next five days, with just a single layer of photoreceptors remaining at P14 (Fig. 5). SNAP-25 removal

affected both photoreceptor outer segments and synaptic terminals (Fig. 7). We conclude here that neuronal SNAP-25 does in fact play essential roles in photoreceptor development and maintenance.

The degeneration observed in our SNAP-25 cKO mice was also worse than other known retinal degeneration mutants, such as those observed in retinal degeneration slow mice (formerly rds, now Prph[Rd2]) which fail to develop outer segments and results in complete loss of peripheral photoreceptors by approximately 9 months of age. CRX knockout mice, which cannot form outer segments and also have impaired synaptogenesis still maintain 2–3 rows of outer nuclear cells at 3 months[53], therefore the degeneration we observed was also faster than CRX knockout mice. The time course of the degeneration we observed is most comparable to that of the retinal degeneration mouse (formerly rd/rd, now Pde6b[rd1]) where development is normal up to P8 before rods begin degenerating[54,55]. Pyknotic nuclei are observed starting at P10 and virtually all rods die off in rd1 mice, however at least 75% of cones still remain at P17. Despite similarities observed between our SNAP-25 cKO mice and rd1 mice, we were able to observe changes in nuclear integrity already by P9, and both rods and cones are equally impacted by SNAP-25 removal. We also confirmed the absence of contamination by Pde6b[rd1] within our colony that could potentially compound with our SNAP-25 cKO and account for the more severe phenotype we observed (Supplementary Fig. 8).

Our results and the recent syntaxin-3 cKO study uncover the unique combination of t-SNARE proteins, neuronal SNAP-25 and syntaxin-3, which play essential roles in photoreceptor development and function. Interestingly, a previous study showed that that omega-3 docosahexaenoic acid increased rhodopsin delivery to the outer segments while enhancing the colocalization of syntaxin-3 and SNAP-25[33]. Additionally, we find that in SNAP-25 cKO mice, photoreceptor degeneration occurred earlier than the degeneration observed in syntaxin-3 cKO mice—even though syntaxin-3 is expressed as early as P2, photoreceptors in syntaxin-3 cKO mice were still abundant throughout the outer nuclear layer at P15[13]. On the other hand, using the same line of CRX-cre mice to conditionally remove SNAP-25 resulted in very few photoreceptors remaining in the outer nuclear layer by P14.

Understanding the processes that underlie photoreceptor trafficking processes is a critical first step in addressing retinal diseases and synaptopathies. Our results show that expression of SNAP-25 is critical for photoreceptor development and maturation and absence of SNAP-25 abolishes vision. One remaining question is whether a different SNAP isoform can take over the role of SNAP-25 at adult ages. It would be interesting for future research to examine whether SNAP-23 or a different SNAP family member can replace the function of SNAP-25 in our SNAP-25 cKO mice. Furthermore, while we did not identify any changes to ERG response following SNAP-23 cKO and do not suspect any regulatory effects are carried out by SNAP-23. However, we do note that these modulatory effects, such as SNAP-23 being a partial fusion agonist, or inhibitory SNARE, have indeed been described before in other systems[26,27]. We also cannot completely rule out that SNAP-25, or a different SNAP isoform, are compensating in the absence of SNAP-23. Since CRX-cre is activated at E12.5 and results in the loss of photoreceptors by the day of eye-opening and precludes the study of a role for SNAP-25 in fully developed photoreceptors. Future studies can take advantage of an inducible cre-recombinase line, such as the CRX-creERT2 line that was described previously[56], to investigate whether SNAP-25 continues to be essential after development. Nevertheless, our research helps further our understanding into the roles of SNARE-mediated exocytosis in non-conventional synapses.

## Methods

**Experimental model and subject details**. SNAP-23 and SNAP-25 flox mice[26,36,37,57–60] of both sexes between postnatal day 4 and up to 1 year of age were used in this study. For the flox mice, loxP sites were introduced surrounding exon 3 of SNAP-23 for SNAP-23 flox mice and loxP sites surrounded exon 5 of SNAP-25 in SNAP-25 flox mice. SNAP-23 and SNAP-25 flox mice mice were crossed with CRX-cre mice (https://www.informatics.jax.org/allele/MGI:5433291) to obtain their respective conditional knockout (cKO) lines. Genomic DNA was obtained from tail clippings using alkaline lysing methods and animals were genotyped using PCR (refer to table for primers). Tail clippings were lysed with 50 mM NaOH and incubated at 96 °C for 1 h with vigorous shaking. Littermate flox/flox mice without CRX-cre were used as controls for all experiments. All mice were maintained on a C57BL/6 background. Our colony was also tested for the presence of the Rd1 mutant (primers listed in Table 1) and is free of contamination by Rd1 (Supplementary Fig. 8).

All mice were housed in a vivarium that was maintained between 22 and 23 °C with a 12-h light on/off cycle. Food and water were accessible ad libitum. We have complied with all relevant ethical regulations for animal use. All experiments detailed here were reviewed and approved by the animal care committee of the University Health Network in accordance with the Canadian Guidelines for Animal Care.

**Preparation of retinal samples for in situ hybridization and immunofluorescence**. Mice were anesthetized with an intraperitoneal injection of sodium pentobarbital (75 mg/kg, Bimeda-MTC) and transcardially perfused with 5–10 ml of PBS followed with 4% PFA. Eyes were placed in 4% PFA following enucleation. Using a 32 G syringe, a hole was poked in the cornea and to aid in the fixation process and eyes were postfixed for 2 h at room temperature in 4% PFA. Following, three PBS washes, the eyes were placed in 30% sucrose overnight at 4 °C for dehydration. The next day, eyes were mounted in OCT and stored at −80 °C until sectioning. Fourteen-μm sections were obtained using a Leica CM3050s cryostat. Sections were dried and stored at −80 °C until use.

**Preparation of retina samples for transmission electron microscopy**. Mice were euthanized as previous describe and transcardially perfused with 5–10 ml of PBS followed by EM fixative. Eyes were enucleated and placed into EM fixative. The front of the eye was removed but the retina and all extra scleral tissue was left intact. An approximately 4 mm × 4 mm square was cut from the mid-periphery of the eye for further processing. Further sample processing for transmission electron microscopy were performed by the Nanoscale Biomedical Imaging Facility, The Hospital for Sick Children, Toronto, Canada. Transmission electron micrographs were taken on a Hitachi HT7800 transmission electron microscope.

**In situ hybridization**. The SNAP-23 probe was generated by introducing a EcoRI site on exon 2 and a BamHI site on exon 8 of SNAP-23 (refer to Table 1 for primers). This fragment of SNAP-23 cDNA was then subcloned into pBluescript SKII. The anti-sense probe was generated using a EcoRI cut and T3 RNA polymerase. The sense probe was generated using a BamHI cut and T7 RNA polymerase. The SNAP-25 probe was generated using SNAP-25 cDNA subcloned into pBluescript SKII. The antisense probe was generated using a BamHI cut and T3 RNA polymerase. The sense probe was generated using a XbaI cut and T7 RNA polymerase. In situ proceeded using an established protocol[44,45]. In short, sections with DIG-labeled probes in hybridization solution were incubated at 65 °C overnight. The following day, 3 stringent washes using solution X (formamide, SSC, SDS) were performed at 72 °C. After 3 TBST washes, sections were blocked in 3% BSA in TBST, then Alkaline phosphatase (AP)-conjugated anti-DIG antibody (Roche) was used 1:2000 to detect the DIG labeled probe. Color reaction was performed the next day using 4-NBT and BCIP in NTMT at room temperature and run until color developed. Following color development, sections were stopped using TE and counterstained with DAPI. Sections were mounted in Mowiol.

**Immunohistochemistry**. Retina sections were washed three for 3 min three times in PBS followed by three by 5-min washes in PBS with 0.1% Triton-X (PBS-Tx). Sections were then blocked with 10% goat serum in PBS-Tx for 1 h. Primary antibody was diluted 1:1000 in blocking solution and left overnight at 4 °C in a humidified chamber. Further antibody detail and information are provided in the attached resource table. The following day, five by 5-min PBS-Tx washes were done, and secondary antibody diluted 1:1000 in with DAPI + PBS-Tx for 1 h at room temperature covered from light. Five more 5-min PBS-Tx washes were done on a shaker covered from light before sections were placed in PBS and mounted in Mowiol.

**Immunofluorescence imaging**. Sections were imaging using Carl Zeiss confocal microscopes and with a minimum averaging of ×4. Images were taken at ×20 and ×63 using 1.0 Airy Unit pinhole sizes. For ×63 oil immersion images, Zeiss immersion oil 518F was used. A region 0.5–1.0 mm away from the optic nerve was selected for imaging and displayed in all representative immunofluorescence images. For S-opsin imaging, the ventral retina was selected for imaging.

**Optical coherence tomography**. Spectral-domain optical coherence tomography scans were performed on mice 2 weeks–1 year of age using the Heidelberg Spectralis animal imaging system. Mice were anesthetized using Avertin (125–250 mg/kg) and had their eyes dilated using Mydriacyl. The left eye of each mouse was used for all scans and all thickness measurements were taken 1.0–1.5 mm away from the optic nerve head, automated segmentation provided by the Heidelberg Spectralis animal imaging software and the experimenter manually fixed erroneous segmentation. Each round of scanning was performed in under 10 min and mice were let to recover on a heated pad.

**Electroretinograms**. Electroretinogram recordings were performed on P30 on SNAP-23 cKO, SNAP-25 cKO and their respective control mice. P16 electroretinograms were also performed on SNAP-25 cKO and control mice. Light stimulation was through the Diagnosys Espion electroretinogram device with a ColorDome Ganzfeld stimulator and Espion software. Before recordings, mice were dark adapted overnight and all recordings were done under red illumination. Mice were anesthetized using Avertin (125–250 mg/kg). Mydriacyl (Alcon) eye drops were used to dilate the eyes, and topical Alcaine (Alcon) was also applied. Body temperature was maintained at 38 °C throughout examination using the Diagnosis device. Recordings were taken from both eyes using gold loop electrodes placed on the cornea of the eyes. Reference electrode was placed between the eyes subcutaneously and a ground electrode in the tail. The cornea was kept moist with a thin layer of methylcellulose solution. Light stimulation consisted of brief pulses of white light (6500 K). Scotopic recordings were done first on the dark-adapted mouse. Ten recordings per stimulus intensity were averaged,

**Table 1 Reagents and resources used in the study.**

| Reagent or resource | Source | Identifier |
|---|---|---|
| Antibodies | | |
| Rabbit monoclonal SNAP-23 | Novus Biologicals | Cat #. NBP2-67157 |
| Mouse monoclonal SNAP-25 | Covance | SMI-81R |
| Rabbit monoclonal SNAP-25 | Abcam | EP3274 |
| Rabbit PSD95 | Abcam | 18258 |
| Mouse PSD95 | Abcam | 2723 |
| Mouse Calbindin D281K | Sigma | CB-955 |
| Mouse PKCα | Santa Cruz | sc-8393 |
| Guinea Pig RIBEYE | Synaptic Systems | 192 104 |
| Mouse Rhodopsin B630 | Novus Bio | NBP2-25160 |
| Rabbit Cone-arrestin | Millipore | 15282 |
| Rabbit S-opsin | Novus Bio | NBP1-20194 |
| Guinea Pig Synaptobrevin-2 | Synaptic Systems | 104 204 |
| Alexa Fluor 488 Goat anti-Rabbit IgG | Invitrogen | A-11008 |
| Alexa Fluor 488 Goat anti-Mouse IgG | Invitrogen | A-11001 |
| Alexa Fluor 568 Goat anti-Rabbit IgG | Invitrogen | |
| Alexa Fluor 568 Goat anti-Mouse IgG | Invitrogen | |
| Alexa Fluor 647 Goat anti-Rabbit IgG | Abcam | |
| Alexa Fluor 647 Goat anti-Mouse IgG | Abcam | |
| Chemicals, peptides, and recombinant proteins | | |
| Taq DNA polymerase | Bio-Helix | Cat #. MB101-0500 |
| Euthanyl | Bimeda-MTC | DIN 00141704 |
| Paraformaldehyde | EM Science | CAS 30525-89-4 |
| PBS Tablets | BioShop | Cat #. PBS404.200 |
| Tissue-Tek O.C.T. Compound | Sakura | Cat #. 62550-12 |
| Triton X-100 | Sigma | CAS 9002-93-1 |
| Goat Serum | Gibco | Cat #. 16210-072 |
| Mowiol 4-88 | Sigma | Cat #. 81381-250 |
| SNAP-25 cDNA | GenScript | OMu17295 |
| Experimental models: organisms/strains | | |
| Mouse: SNAP23[tm1Jpes] (SNAP-23 flox) | OZGene, Feng et al.[27] | N/A |
| Mouse: B6-Snap25[tm3mcw] (SNAP-25 flox) | [57] | |
| Mouse: Tg(Crx-cre)[764Gla] (CRX-cre) | [38] | |
| Mouse: SNAP23[tm1Jpes] ; Tg(Crx-cre)[764Gla] (SNAP-23 cKO) | This paper | N/A |
| Mouse: B6-Snap25[tm3mcw]; Tg(Crx-cre)[764Gla] (SNAP-25 cKO) | This paper | N/A |
| Oligonucleotides | | |
| 5′-GGGGGTGAGTTGAAGTCATTGAAG-3′ | [37] | SNAP-23 (Forward) |
| 5′-AGCTTAAACGGGATGAACTCAGGC-3′ | [37] | SNAP-23 (Reverse) |
| 5′-CCCTGGGGAACCACGGCAGA-3′ | This paper | SNAP-25 (Forward) |
| 5′-TCCCAGGAAACAGCACAGCGT-3′ | This paper | SNAP-25 (Reverse) |
| 5′-AAACCTGAAGCAGGATGTGAGT-3′ | This paper | (1919) Cre (Forward) |
| 5′-ACAGAAGCATTTTCCAGGTATGCT-3′ | [37] | (1747) Cre (Reverse) |
| 5′-CTACAGCCCCTCTCCAAGGTTTATAG-3′ | [61] | Common (Forward) |
| 5′-ACCTGCATGTGAACCCAGTATTCTATC-3′ | [61] | WT (Reverse) |
| 5′-TCG CTG ATC CTT GGG AGG GTC TC-3′ | Jax Protocol 31415 | RD1 mutant (Reverse) |
| 5′-GAGAATTCACCATGGATAATCTGTCCCCAGA-3′ | This paper | SNAP-23 in situ probe EcoRI |
| 5′-GCGGATCCTTAACTATCAATGAGTTTCTTT-3′ | This paper | SNAP-23 in situ probe BamHI |
| Software and algorithms | | |
| Adobe Photoshop | Adobe | www.adobe.com |
| Adobe Illustrator | Adobe | www.adobe.com |
| ImageJ | National Institute of Health | imagej.net/software/fiji |
| Origin Pro 2016 | OriginLab | www.originlab.com/origin |
| Prism 9.5.1 | GraphPad | www.graphpad.com |
| Espion | Diagnosys | diagnosysllc.com |
| Spectralis | Heidelberg Spectralis | heidelbergengineering.com |
| OptoMotry | CerebralMechanics | www.cerebralmechanics.com |
| Zen Blue | Carl Zeiss | zeiss.com |
| Other | | |
| Microscope Cover Glass | Fisherbrand | Cat #. 12545M |
| Superfrost Plus Slides | Fisherbrand | Cat #. 12-550-15 |

and the a- and b-wave amplitudes were manually determined by the experimenter. Scotopic stimuli conditions are as follows: 0.00025, 0.0025, 0.025, 0.25, 2.5, 5, and 10 cd/s/m$^2$. Mice were then light adapted with 30 cd/s/m$^2$ background illumination for 10 min. Photopic tests were done in the presence of 30 cd/s/m$^2$ background illumination. Ten recordings per flash intensity were averaged, and the amplitude of the a- and b-waves was manually determined by the experimenter again. The following photopic stimulus intensities were used: 0.15, 1.25, 5, 10, 25, 50, and 75 cd/s/m$^2$.

**Optomotry testing**. Quantification of visual acuity was performed using the OptoMotry system from CerebralMechanics. Vertical sine wave gratings were presented by a chamber composed of four monitors to create a virtual cylinder in the visual field of the mouse. A Zeiss video camera was used to image the mouse from above. The mouse's head was manually tracked by a software and the vertical sine wave gratings were then rotated around the mouse at 12 degrees/s using the randomized protocol design and the simple staircase psychophysical method within the software. The tester was blinded to spatial frequencies being tested during the procedure.

**Statistics and reproducibility**. All statistical analyses were done in OriginPro2016 and Prism 9.5.1. Independent $t$ tests were used for two-group experiments with a $p$ value $< 0.05$ as the threshold for statistical significance. For all quantified tests (OCT, ERG, OKT), minimum triplicates were used at every time point. No data were excluded from analysis. Reproducibility was confirmed using triplicates. Two-way analysis of variance (ANOVA) was used for comparisons of multiple groups, with a significance level of 0.05. All error bars represent SEM.

**Reporting summary**. Further information on research design is available in the Nature Portfolio Reporting Summary linked to this article.

## Data availability

Source data for all graphs and charts presented can be found in Supplementary Data 1. Further information and requests for resources and reagents should be directed to and will be fulfilled by the lead contact, Shuzo Sugita (shuzo.sugita@uhnresearch.ca).

## Materials availability

All reagents and mouse lines generated by this study are available from the lead contact.

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

## Acknowledgements
We thank Weichun Lin (UT Southwestern) for critically reading an earlier version of our manuscript and for kindly providing SNAP-25 flox mice. This work was supported by the Natural Sciences and Engineering Research Council of Canada (RGPIN 2020 07139) and the Canadian Institutes of Health Research (CIHR PJT 165917). M.H. and C.H.C. are supported by the Vision Science Research Program scholarship, Canada Graduate Scholarship, and the Ontario Graduate Scholarship. The authors wish to thank the Nanoscale Biomedical Imaging Facility, The Hospital for Sick Children, Toronto, Canada for assistance with transmission electron micrographs.

## Author contributions
M.H.: methodology, investigation, writing—original draft, review, and editing, visualization. C.H.C.: methodology, investigation, writing—review and editing, visualization. A.G.: methodology, investigation, writing—review and editing. H.H. and V.Q.B.P.T.: methodology, investigation. S.E., H.-S.S., Z.-P.F., P.P.M.: resources, writing—review and editing. V.A.W.: conceptualization, resources, writing—review and editing. S.S.: conceptualization, writing—original draft, supervision.

## Competing interests
The authors declare no competing interests.
