## [Peer Review File · Communications Biology]

Reviewers' comments:

Reviewer #1 (Remarks to the Author):

The study by Huang and colleagues on SNAP25 and photoreceptor function and survival in mice is a timely study that describes the consequences of SNAP25 gene inactivation selectively in rod and cone photoreceptors in the developing mouse retina. The authors utilize multiple approaches, both visual and functional, to assess the retina. The authors report that inactivation of the SNAP25 gene, but not the SNAP23 gene, leads to the rapid degeneration of photoreceptors in the early postnatal period, shortly after they have matured. Overall, this study has the potential to add to our understanding of the roles that SNARE proteins have in photoreceptor function and survival. However, the text requires significant revision, and additional information and quantification are needed in order to more fully appreciate the findings.

Specific comments

1. That SNAP25 is present in photoreceptors is no longer really in contention at this time. In addition, multiple groups have shown that SNAP25 interacts with the isoform of syntaxin 3 that is expressed in the retina, and SNAP25 has been previously implicated in photoreceptor outer segment trafficking. Therefore, statements in the text regarding a controversy about whether SNAP-25 is utilized by photoreceptors need to be significantly revised or removed.

A second source of confusion is that the authors expected that syntaxin 3 would not interact with SNAP25. They seem to have erroneously equated retinal syntaxin 3 with the ubiquitous syntaxin 3, despite evidence from multiple studies indicating that the syntaxin 3 isoform expressed in the retina is NOT the ubiquitously expressed syntaxin 3A. In contrast to syntaxin 3A, the retinal syntaxin 3 isoform does bind SNAP25.

The authors are strongly encouraged to review the relevant literature related to SNAP25 and also to retinal syntaxin 3, particularly from the labs of Morgans, Naash, Deretic, and Janz, and to correct these misrepresentations, beginning with the opening statement of the abstract.

2. When observing photoreceptor degeneration, it is important to note whether the mice under investigation are free of common Rd mutations, including the Rd1 mutation. Have the authors screened for common Rd mutations in their lines? This is important to do as some mutations are not apparent in some mouse strains, yet could potentially enhance degeneration in the presence of another mutation. An occult Rd mutation might explain the more severe phenotype of SNAP25 cKO relative to retinal syntaxin 3 cKO. Also with respect to the mice, authors should specifically state the background strain of the animals used in their experiments in the Methods section.

3. The authors assert that photoreceptors are almost completely absent by P14. In Figure 4, the ERGs were performed at P30 and in the supplement on P16. The authors conclude "that in the absence of SNAP-25, photoreceptors are neither hyperpolarizing at the onset of light stimulation, nor passing on the light signal." That is because there are no photoreceptors. One cannot interpret the lack of a and b waves in the manner that the authors have done. Minimally, one would need to perform the ERG studies on at a much earlier stage, prior to P14, when photoreceptors were still present, to determine whether or not photoreceptors lacking SNAP-25 are responsive to light or can pass on the light signal. Furthermore, the authors would also need to know the state of the photoreceptor outer segments. This sentence must be revised or removed.

4. Figure 8. When comparing S-opsin labeling, it is important to know whether one is looking at the same region of the retina in both panels, as the distribution changes rather dramatically depending on where one is looking in the retina. In what region of the retina were these images taken? Where both samples obtained from a similar region of the retina? If the regions from which the retinal sections

were obtained are not known then this would cast doubt as to the reliability of the comparisons shown in this Figure and others. Information about retinal location should be included in Methods.

5. It is not clear why bipolar cells are not included in the list of cells that show SNAP-25 expression in Figure 3. How was SNAP25 expression in bipolar cells ruled-out? Certainly the pattern of immunolabeling for SNAP25 in subsequent figures is suggestive of bipolar cells. Please explain.

6. For all figures, the number of animals that were examined for each experimental condition should be included to provide the reader with an indication of the reproducibility of the findings. In addition, images from more than one animal per condition should be shown (or included as supplemental material) to reduce concerns about sample bias when showing representative figures. An alternative would be to provide quantification across multiple animals.

7. Figure 7 – In panel B, the region below the synaptic boundary looks distorted in the SNAP25 cKO animals. Was this a common finding and if so, what is the interpretation? If not, what is different about the sample shown and what would a more representative sample show? In panel C, are spherical ribbons ever observed in the control animals? If the authors have examined multiple samples, and never see spherical ribbons in the control samples, this would be worth noting, as would the percentage of ribbons observed that are spherical for control and cKO animals.

8. Figure 5C shows that in the P14 cKO of SNAP25, the ONL is still present, with photoreceptor somata showing SNAP25 labeling, and there are ribbons in the IPL. Are the figure panels labeled correctly?

Minor Comments

1. "Further examination into why photoreceptors utilize such a unique combination of t-SNAREs could give important insights and explain why the rates of degeneration vary between these two SNARE proteins." The meaning of this sentence is not quite clear. Please revise.

2. In listing locations where ribbon synapses are found, the authors could also include vestibular hair cells.

Reviewer #2 (Remarks to the Author):

Vesicle fusion in neurons depends on the formation of a SNARE complex that is formed by a v-SNARE protein, synaptobrevin, and two t-SNARE proteins, typically syntaxin-1 and SNAP25. Photoreceptors have an unusual t-SNARE protein, syntaxin-3, and SNAP25 has been previously difficult to detect in photoreceptors. Therefore, it has been previously speculated that SNAP23, instead of SNAP-25, could be the other t-SNARE protein in photoreceptors.

In the paper by Huang and colleagues, the authors revisited this important question analyzed whether the t-SNARE protein SNAP23 or the t-SNARE protein SNAP25 is the most relevant for photoreceptors in the mouse retina. For this purpose, they generated photoreceptor-specific knockout mice for SNAP23 and SNAP25 and analyzed the effects of this deletion with a broad spectrum of techniques. The applied techniques include morphological analyses (OCT, in-situ-hybridization, RNAscope, immunofluorescence microscopy with various photoreceptor markers in developing and mature retina, electron microscopy) functional analyses (ERG analyses, optometry testing).

Using these methods, the authors found that photoreceptor-specific deletion of SNAP-23 has very little, if any effect, on photoreceptor morphology and function whereas photoreceptor-specific deletion of SNAP25 has very strong effects both on photoreceptor morphology and photoreceptor function. These SNAP25-dependent functions in photoreceptors were consistently shown by the morphological analyses as well as by the functional analyses. That was the case both for the continuously active ribbon synapses as well as for the inner/outer segments needed for phototransduction. These data

clearly show that SNAP25, but not SNAP23, is the relevant t-SNARE in photoreceptors.

The experiments were expertly performed and convincingly documented. The data are very consistent and scientifically very important. The paper is well written. I only have a few points of criticism that should be addressed in a revised version of the manuscript.

Points of criticism

1.) Introduction, page 3, first paragraph:

"Photons of light activate opsin pigments starting the phototransduction cascade using oxidation and reoxidation reactions." This is not an appropriate summary of the phototransduction cascade but summarizes only its toxic side reactions. Please re-phrase.

2.) Results, page 5, first paragraph: Do the authors really mean "post-translational modifications to mRNA" or do they mean post-transcriptional?

3.) Results, page 7, last paragraph "In strong contrast", instead of "In stark contrast".

4.) Fig. 2A no units were given for the Y-axis in Fig. 2A, also not for the Y-axis in Fig. 4 C, E. Please add.

5.) Results, Fig 5A,C versus Fig. 7. In Fig. 5A the authors observed a loss of RIBEYE fluorescence, but in the EM they showed that synaptic ribbons are present in the photoreceptor synapses. Isn't that a contradiction? Previous knockout analyses of RIBEYE have demonstrated that absence of RIBEYE leads to a complete disappearance of synaptic ribbons. Please comment.

6.) The retinal layers in Fig. 5A, B, and C should be labeled.

7.) In Fig. 5A, P14, there is a substantial SNAP25 immunosignal left in the photoreceptor somata in the ONL. Can the authors please comment on that. Is it a Cre-less region or does it represent SNAP25 e.g., in Müller glia cells?

8.) Fig. 5C is confusing. I assume, the images were mixed/mislabeled. In the upper right panel of Fig. 5C: the presented image does not appear to come from the photoreceptor-specific knockout because it shows a strong SNAP25 immunosignal in the ONL. I think the upper two panels arise from the control retina and the lower two panels from the photoreceptor-specific knockout. Most likely, the labeling of the figure, as it is, is wrong. Furthermore, in the images from the photoreceptor-specific knockout, also the ribbons in the inner plexiform layer (IPL) appear to be absent. This is surprising because the IPL contains the synapses between bipolar cells and ganglion cells that are not affected by the knockout. Can the authors comment on that?

9.) In Fig. 5A-C, the sclera is also substantially labelled by SNAP25 antibody, as well as by PSD95 antibody. The authors might want to explain this in the figure legend.

10) Fig 7: Please add a higher magnification electron micrograph of the inner/outer segment region from the knockout and the control retina to allow for better comparing potential changes in the ultrastructure. This is very difficult with the low power magnification electron micrographs given in Fig 7A.

Reviewer #3 (Remarks to the Author):

The article authored by Huang and colleagues, titled "SNAP-25, but not SNAP-23, is essential for photoreceptor function and survival in mice" addresses the ongoing debate surrounding the functions of SNAP-23 and SNAP-25 in photoreceptors. The authors carried out a comprehensive reevaluation of SNAP-23 and SNAP-25 expression in the retina, conducting experiments involving conditional knockout (cKO) mice in which SNAP23 and SNAP25 were specifically deleted from photoreceptor cells to investigate their respective roles. Their investigations encompassed techniques such as immunohistochemistry, electrophysiology, and transmission electron microscopy. These experiments unveiled that the deletion of SNAP-23 did not seem to have any noticeable impact on retinal morphology or visual function.

In the case of photoreceptors lacking SNAP-25, they exhibited what appeared to be normal development until postnatal day 9 (P9), after which a significant proportion underwent degeneration

by P14. This degeneration resulted in severe retinal thinning and a profound loss of vision. The loss of photoreceptors in SNAP-25 cKO mice was linked to the suppression of electroretinograms and a complete loss of visual function. In summary, their findings indicate that SNAP-25 plays a pivotal role in facilitating the transport of photopigments and synaptic function—essential processes for the health and survival of photoreceptors.

Major comments:

1. Typically, conditional knockout mice are generated to investigate the function of the gene of interest in either the developmental phase, the mature stages, or both.

However, in this particular situation, this conventional approach was not followed. Instead, it introduced complexity into the scenario, which is regrettable. There is a complete disparity between the title and the outcomes presented in the article. Towards the conclusion of the article, readers might become perplexed because important questions and suggestions have not been addressed clearly. Is snap25 indispensable for development, survival, and photoreceptor function?

2. The article does not provide any insights into the kinetics of CRE expression in the SNAP25 cKO, leaving the reader unaware of when the inactivation of the SNAP-25 gene begins. This is a critical aspect to distinguish between developmental and functional effects.

The normal development of photoreceptors until P9 could potentially be attributed to the presence of a pre-existing reserve pool before the onset of SNAP-25 inactivation.

The observed phenomenon appears to be developmental in nature, as there are early disruptions in membrane trafficking, leading to a variety of effects. To establish SNAP-25's involvement in photoreceptor function, it may be advisable to delete SNAP-25 at an adult stage.

3. Given the sub-millisecond timeframes of photoreceptor synaptic transmission, suggesting that SNAP23 could fulfill this function appears inappropriate. Furthermore, there is a lack of data to confirm that SNAP23 is not essential for photoreceptor function, as it's likely that its absence can be compensated for by SNAP25 or other isoforms.

4. The authors have made only a limited effort to establish connections between the synapses of photoreceptors and auditory sensory cells. Notably, they have overlooked the recent work by Calvet et al. (PMID 36483015), which offers fresh insights into the function of SNAP25 in these cells. This omission is unfortunate because it could have enriched both results and discussion within the current article. In the realm of research, keeping updated to date is not just important but necessary to progress.

Minor comments

1. The clarity of the writing, as well as the quality of the figures and their legends, can be enhanced to better facilitate the reader's understanding. Each immunofluorescence panel should incorporate high-magnification views to emphasize the authors' assertions.

2. it would be helpful to mention the retinal region in all figures.

Reviewer #1:

The study by Huang and colleagues on SNAP25 and photoreceptor function and survival in mice is a timely study that describes the consequences of SNAP25 gene inactivation selectively in rod and cone photoreceptors in the developing mouse retina. The authors utilize multiple approaches, both visual and functional, to assess the retina. The authors report that inactivation of the SNAP25 gene, but not the SNAP23 gene, leads to the rapid degeneration of photoreceptors in the early postnatal period, shortly after they have matured. Overall, this study has the potential to add to our understanding of the roles that SNARE proteins have in photoreceptor function and survival. However, the text requires significant revision, and additional information and quantification are needed in order to more fully appreciate the findings.

- We thank the reviewer for seeing the potential of our manuscript.

Specific comments

1. That SNAP25 is present in photoreceptors is no longer really in contention at this time. In addition, multiple groups have shown that SNAP25 interacts with the isoform of syntaxin 3 that is expressed in the retina, and SNAP25 has been previously implicated in photoreceptor outer segment trafficking. Therefore, statements in the text regarding a controversy about whether SNAP-25 is utilized by photoreceptors need to be significantly revised or removed.

A second source of confusion is that the authors expected that syntaxin 3 would not interact with SNAP25. They seem to have erroneously equated retinal syntaxin 3 with the ubiquitous syntaxin 3, despite evidence from multiple studies indicating that the syntaxin 3 isoform expressed in the retina is NOT the ubiquitously expressed syntaxin 3A. In contrast to syntaxin 3A, the retinal syntaxin 3 isoform does bind SNAP25.

The authors are strongly encouraged to review the relevant literature related to SNAP25 and also to retinal syntaxin 3, particularly from the labs of Morgans, Naash, Deretic, and Janz, and to correct these misrepresentations, beginning with the opening statement of the abstract.

- We thank the reviewer's thoughtful suggestion about our literature review. We have now cited more literature that supports the expression of SNAP-25 in photoreceptors as well as its interaction with photoreceptor syntaxin-3B (lines 108-111). At the same time, there is a relatively recent review on retina synaptogenesis which still reports on the conflict of the expression of SNAP-25 in photoreceptors (Burger et al., 2021, Dev. Biol). Nevertheless, we tuned down our statements and our framing regarding the controversy, focusing more on the unknown functions rather than expression in our abstract and introduction. We also rewrote a more balanced literature review and cited more relevant literature (lines 98-115). We hope that this is satisfactory.

2. When observing photoreceptor degeneration, it is important to note whether the mice under investigation are free of common Rd mutations, including the Rd1 mutation. Have the authors screened for common Rd mutations in their lines? This is important to do as some mutations are not apparent in some mouse strains, yet could potentially enhance degeneration in the presence

of another mutation. An occult Rd mutation might explain the more severe phenotype of SNAP25 cKO relative to retinal syntaxin 3 cKO. Also with respect to the mice, authors should specifically state the background strain of the animals used in their experiments in the Methods section.

- We have now clarified in the methods section the background our mice are being maintained on (C57BL/6). While we have not screened for all known mutations, we note that our SNAP-25 colony exhibits phenotypes as expected; ie. all Cre⁺ mice exhibit a phenotype while Cre negative control mice do not. We independently maintain a pure flox/flox colony of which all mice did not exhibit any degenerative phenotypes. However, as suggested by the reviewer, we tested for the well-known RD1 mutation in our present colony, and of mice analyzed within the manuscript and attached the results as a supplement (supplemental Figure 8). The Rd1 mutant band (115 bp) is present in our Rd1 positive control, however, this mutant band is not present within our current colony or in mice used for analysis.

3. The authors assert that photoreceptors are almost completely absent by P14. In Figure 4, the ERGs were performed at P30 and in the supplement on P16. The authors conclude “that in the absence of SNAP-25, photoreceptors are neither hyperpolarizing at the onset of light stimulation, nor passing on the light signal.” That is because there are no photoreceptors. One cannot interpret the lack of a and b waves in the manner that the authors have done. Minimally, one would need to perform the ERG studies on at a much earlier stage, prior to P14, when photoreceptors were still present, to determine whether or not photoreceptors lacking SNAP-25 are responsive to light or can pass on the light signal. Furthermore, the authors would also need to know the state of the photoreceptor outer segments. This sentence must be revised or removed.

- We only performed ERG experiments at P16 after the loss of the majority of photoreceptors, hence, our statement of “in the absence of SNAP-25, photoreceptors are neither hyperpolarizing at the onset of light stimulation, nor passing on the light signal” was indeed not experimentally supported. Accordingly, we have now removed this sentence to prevent any confusion.

4. Figure 8. When comparing S-opsin labeling, it is important to know whether one is looking at the same region of the retina in both panels, as the distribution changes rather dramatically depending on where one is looking in the retina. In what region of the retina were these images taken? Where both samples obtained from a similar region of the retina? If the regions from which the retinal sections were obtained are not known then this would cast doubt as to the reliability of the comparisons shown in this Figure and others. Information about retinal location should be included in Methods.

- Thank you for the important comment. For all S-opsin stains, a region from the ventral retina 0.5 mm – 1.0 mm away from the optic nerve was selected for imaging to account for differences in S-opsin distribution. We apologize that this important detail was omitted from our first submission. Further, to ensure reliability, we’ve repeated the S-opsin stains in two more independent samples and quantified the S-opsin puncta. For quantifications, n = 3 control and n = 3 SNAP-25 cKO mice aged within 24 hours of postnatal day 9 were used. 4 images were

quantified from each mouse. The quantifications and details are now appended to Figure 8, the figure legend, and methods.

Response letter Figure 1. Updated figure 8.

We have modified figure 8 to include panel 8C, quantifications of S-opsin.

5. It is not clear why bipolar cells are not included in the list of cells that show SNAP-25 expression in Figure 3. How was SNAP25 expression in bipolar cells ruled-out? Certainly the pattern of immunolabeling for SNAP25 in subsequent figures is suggestive of bipolar cells. Please explain.

- In our legend to Figure 3, we omitted bipolar cells initially to reflect earlier publications that found SNAP-25 in horizontal cells and amacrine cells (Hirano et. al., 2011, J Comp Neurol.).

However, we agree that the immunofluorescence data of SNAP-25 does suggest the presence of SNAP-25 in bipolar cells. To confirm, we co-stained for SNAP-25 and bipolar cells using SNAP-25 SMI81 and Chx-10 antibodies respectively. We find that SNAP-25 immunofluorescence indeed surrounds bipolar cells (Response letter Figure 2), and have now updated our text and figure legends to reflect this.

Response letter Figure 2. SNAP-25 is found in bipolar cells in addition to horizontal and amacrine cells. When SNAP-25 SMI81 (green) is costained with Chx-10 for bipolar cells (red), we find that SNAP-25 indeed surrounds Chx-10 bipolar cells.

6. For all figures, the number of animals that were examined for each experimental condition should be included to provide the reader with an indication of the reproducibility of the findings. In addition, images from more than one animal per condition should be shown (or included as supplemental material) to reduce concerns about sample bias when showing representative figures. An alternative would be to provide quantification across multiple animals.

- We have updated figure legends with the number of animals that were examined for each experiment. We've also added quantifications to S-opsin stains represented in Figure 8 to help ensure reproducibility for stains with regional differences to avoid sample bias.

7. Figure 7 – In panel B, the region below the synaptic boundary looks distorted in the SNAP25 cKO animals. Was this a common finding and if so, what is the interpretation? If not, what is different about the sample shown and what would a more representative sample show? In panel C, are spherical ribbons ever observed in the control animals? If the authors have examined multiple samples, and never see spherical ribbons in the control samples, this would be worth noting, as would the percentage of ribbons observed that are spherical for control and cKO animals.

- The distortion under the synaptic boundary in SNAP-25 cKO was a common finding, and we interpret this to be the sprouting of ectopic synapses. We have made a note of this in the text (lines 252-254). Spherical ribbons were indeed observed in the control animals but to a lesser extent. Due to the nature of spherical ribbons being a feature of developing ribbon synapses, to increase the appreciation of our findings, we quantified 70+ synaptic ribbons according to their state as described previously (Regus-Leidig, 2009) in supplemental Figure 7 and discussed on lines 258-261. As expected, we find that spherical ribbons represent 27% of control ribbons and 38% of SNAP-25 cKO ribbons, suggesting SNAP-25 cKO contains more spherical ribbons.

8. Figure 5C shows that in the P14 cKO of SNAP25, the ONL is still present, with photoreceptor somata showing SNAP25 labeling, and there are ribbons in the IPL. Are the figure panels labeled correctly?

- We apologize for swapping the figures – the higher magnification image of the control was placed in the SNAP-25 cKO column which is why SNAP-25 remained in the photoreceptors shown in the SNAP-25 cKO column. We have fixed this mistake now.

Response letter Figure 3. Updated figure 5.

We have added labels for retinal layers across panels A-C. We also fixed the swapped panels of figure 5C, where we originally placed high magnification control images under the SNAP-25 cKO column and low magnification SNAP-25 cKO images under the control panel.

Minor Comments

1. “Further examination into why photoreceptors utilize such a unique combination of t-SNAREs could give important insights and explain why the rates of degeneration vary between these two SNARE proteins.” The meaning of this sentence is not quite clear. Please revise.

- We have removed this confusing sentence.

2. In listing locations where ribbon synapses are found, the authors could also include vestibular hair cells.

- We have now added vestibular hair cells as a location ribbon synapses are found (line 76, line 298).

Reviewer #2:

Vesicle fusion in neurons depends on the formation of a SNARE complex that is formed by a v-SNARE protein, synaptobrevin, and two t-SNARE proteins, typically syntaxin-1 and SNAP25. Photoreceptors have an unusual t-SNARE protein, syntaxin-3, and SNAP25 has been previously difficult to detect in photoreceptors. Therefore, it has been previously speculated that SNAP23, instead of SNAP-25, could be the other t-SNARE protein in photoreceptors.

In the paper by Huang and colleagues, the authors revisited this important question analyzed whether the t-SNARE protein SNAP23 or the t-SNARE protein SNAP25 is the most relevant for photoreceptors in the mouse retina. For this purpose, they generated photoreceptor-specific knockout mice for SNAP23 and SNAP25 and analyzed the effects of this deletion with a broad spectrum of techniques. The applied techniques include morphological analyses (OCT, in-situ-hybridization, RNAscope, immunofluorescence microscopy with various photoreceptor markers in developing and mature retina, electron microscopy) functional analyses (ERG analyses, optometry testing).

Using these methods, the authors found that photoreceptor-specific deletion of SNAP-23 has very little, if any effect, on photoreceptor morphology and function whereas photoreceptor-specific deletion of SNAP25 has very strong effects both on photoreceptor morphology and photoreceptor function. These SNAP25-dependent functions in photoreceptors were consistently shown by the morphological analyses as well as by the functional analyses. That was the case both for the continuously active ribbon synapses as well as for the inner/outer segments needed for phototransduction. These data clearly show that SNAP25, but not SNAP23, is the relevant t-SNARE in photoreceptors.

The experiments were expertly performed and convincingly documented. The data are very consistent and scientifically very important. The paper is well written. I only have a few points of criticism that should be addressed in a revised version of the manuscript.

- We appreciate the reviewer's kind evaluation of our manuscript and the importance of our work. We are happy about this positive feedback from the reviewer.

Points of criticism

1.) Introduction, page 3, first paragraph:

“Photons of light activate opsin pigments starting the phototransduction cascade using oxidation and reoxidation reactions.” This is not an appropriate summary of the phototransduction cascade but summarizes only its toxic side reactions. Please re-phrase.

- We have altered this paragraph to detail the relevant discussion on the phototransduction process (lines 62-66).

2.) Results, page 5, first paragraph: Do the authors really mean “post-translational modifications to mRNA” or do they mean post-transcriptional?

- The reviewer is correct, we meant post-transcriptional. We thank the reviewer for catching this mistake.

3.) Results, page 7, last paragraph “In strong contrast”, instead of “In stark contrast”.

- We have made this change (line 216).

4.) Fig. 2A no units were given for the Y-axis in Fig. 2A, also not for the Y-axis in Fig. 4 C, E. Please add.

- We have now updated the panels to include the stimulation intensity units (cd/s/m²).

5.) Results, Fig 5A,C versus Fig. 7. In Fig. 5A the authors observed a loss of RIBEYE fluorescence, but in the EM they showed that synaptic ribbons are present in the photoreceptor synapses. Isn't that a contradiction? Previous knockout analyses of RIBEYE have demonstrated that absence of RIBEYE leads to a complete disappearance of synaptic ribbons. Please comment.

- Represented in Figure 5A is PSD95 immunofluorescence which stains for the photoreceptor presynaptic terminals and is not specific to synaptic ribbons, we have updated the text to make this more clear (line 222). While Figure 5C does show a loss of RIBEYE fluorescence, we would like to highlight that the mice used in Figure 5C are P14, while the EM represented in Figure 7 was done at P9. A more accurate comparison would be between the EM shown in Figure 7 and the RIBEYE immunohistology depicted in Figure 8b where mice are at a comparable age (P9 for both). We do not believe the presence of synaptic ribbons in EM at P9 to be a contradiction of our earlier results since the presence of RIBEYE staining can still be observed by immunofluorescence at P9. To increase clarity, we've updated the figures and figure legends so experimental ages are more explicitly stated in both.

6.) The retinal layers in Fig. 5A, B, and C should be labeled.

- Figure 5 has now been updated with labels for retinal layers.

7.) In Fig. 5A, P14, there is a substantial SNAP25 immunosignal left in the photoreceptor somata in the ONL. Can the authors please comment on that. Is it a Cre-less region or does it represent SNAP25 e.g., in Müller glia cells?

- Perhaps the lack of labeling of the retinal layers created confusion on the area of SNAP-25 staining, we do not believe there to be substantial SNAP-25 signal remaining in P14 photoreceptors. The weak SNAP-25 immunofluorescent signal is coming from ectopic sprouting of the postsynapses. As a fix, we have now added labels and pointed out instances where the postsynapse ectopically extends into the outer nuclear layer and may contribute to the perceived ONL SNAP-25 signal in the magnified panels.

8.) Fig. 5C is confusing. I assume, the images were mixed/mislabeled. In the upper right panel of Fig. 5C: the presented image does not appear to come from the photoreceptor-specific knockout because it shows a strong SNAP25 immunosignal in the ONL. I think the upper two panels arise from the control retina and the lower two panels from the photoreceptor-specific knockout. Most likely, the labeling of the figure, as it is, is wrong. Furthermore, in the images from the photoreceptor-specific knockout, also the ribbons in the inner plexiform layer (IPL) appear to be absent. This is surprising because the IPL contains the synapses between bipolar cells and ganglion cells that are not affected by the knockout. Can the authors comment on that?

- We apologize for the swapped figures, as the reviewer pointed out the upper two panels were from the control mouse while the lower two panels from our SNAP-25 cKO. We appreciate the reviewer for going through our figures meticulously. We have fixed this now and added labels for the retinal layers to the figures. Regarding the ribbons in the inner plexiform layer, this is a limitation of our RIBEYE antibody as it does not discretely pick-up the much smaller bipolar cell ribbons found in the IPL (Sterling and Matthew, 2005). Therefore, we do not believe our photoreceptor-specific knockout is affecting the ribbon synapses of bipolar cells.

9.) In Fig. 5A-C, the sclera is also substantially labelled by SNAP25 antibody, as well as by PSD95 antibody. The authors might want to explain this in the figure legend.

- We have now noted this in the figure legend.

10) Fig 7: Please add a higher magnification electron micrograph of the inner/outer segment region from the knockout and the control retina to allow for better comparing potential changes in the ultrastructure. This is very difficult with the low power magnification electron micrographs given in Fig 7A.

- Due to the nature of the developing inner/outer segment region, we elected to image so both regions were present within the same image, posing a restriction to image at higher magnifications. However, we agree that the micrographs presented in Figure 7A are indeed small and have added two larger panels at 6000x magnification in our supplementals (supplemental Figure 6). We hope the high magnification images will provide more details regarding the retinal structures supporting the conclusion made in Figure 7A.

Reviewer #3:

The article authored by Huang and colleagues, titled "SNAP-25, but not SNAP-23, is essential for photoreceptor function and survival in mice" addresses the ongoing debate surrounding the functions of SNAP-23 and SNAP-25 in photoreceptors. The authors carried out a comprehensive reevaluation of SNAP-23 and SNAP-25 expression in the retina, conducting experiments involving conditional knockout (cKO) mice in which SNAP23 and SNAP25 were specifically deleted from photoreceptor cells to investigate their respective roles. Their investigations encompassed techniques such as immunohistochemistry, electrophysiology, and transmission electron microscopy. These experiments unveiled that the deletion of SNAP-23 did not seem to have any noticeable impact on retinal morphology or visual function.

In the case of photoreceptors lacking SNAP-25, they exhibited what appeared to be normal development until postnatal day 9 (P9), after which a significant proportion underwent degeneration by P14. This degeneration resulted in severe retinal thinning and a profound loss of vision. The loss of photoreceptors in SNAP-25 cKO mice was linked to the suppression of electroretinograms and a complete loss of visual function. In summary, their findings indicate that SNAP-25 plays a pivotal role in facilitating the transport of photopigments and synaptic function—essential processes for the health and survival of photoreceptors.

- We appreciate the reviewer for succinctly summarizing and capturing the essence of our work.

Major comments:

1. Typically, conditional knockout mice are generated to investigate the function of the gene of interest in either the developmental phase, the mature stages, or both. However, in this particular situation, this conventional approach was not followed. Instead, it introduced complexity into the scenario, which is regrettable. There is a complete disparity between the title and the outcomes presented in the article. Towards the conclusion of the article, readers might become perplexed because important questions and suggestions have not been addressed clearly. Is snap25 indispensable for development, survival, and photoreceptor function?

- We recognize that not providing information regarding the expression profile of CRX-cre may lead to readers to infer that our recombination only occurs after photoreceptor development at P9. We would like to emphasize that CRX-cre is active throughout postnatal development (Kakakhel et. al., 2020, supplemental figure 2). To this end, we have now revised our text to include information about when our cre-recombinase becomes active and provided the appropriate citations (lines 142, 357-359). We have also modified our title to reflect the referee's questions, that SNAP-25 is essential for photoreceptor development, survival, and function, which we believe are indeed answered within the findings of this manuscript.

2. The article does not provide any insights into the kinetics of CRE expression in the SNAP25 cKO, leaving the reader unaware of when the inactivation of the SNAP-25 gene begins. This is a critical aspect to distinguish between developmental and functional effects.

The normal development of photoreceptors until P9 could potentially be attributed to the presence of a pre-existing reserve pool before the onset of SNAP-25 inactivation.

The observed phenomenon appears to be developmental in nature, as there are early disruptions in membrane trafficking, leading to a variety of effects. To establish SNAP-25's involvement in photoreceptor function, it may be advisable to delete SNAP-25 at an adult stage.

- CRX-cre is expressed starting embryonic day E12.5 and throughout postnatal development as described previously (Prasov and Glaser, 2012, Kakakhel. et. al., 2020). We have now added information regarding CRX-cre kinetics in the text so readers are aware that SNAP-25 is absent in photoreceptors during development and not only removed after P9 (lines 142, 357-359). We also agree that one limitation to our study lies in that photoreceptors are dead by adulthood and a role for SNAP-25 in fully developed photoreceptors cannot be accessed. We now include a paragraph in our discussion regarding future directions and the possibility of using inducible CRX-creERT2 in adult mice to address a role of SNAP-25 in developed photoreceptors (lines 359-361).

3. Given the sub-millisecond timeframes of photoreceptor synaptic transmission, suggesting that SNAP23 could fulfill this function appears inappropriate. Furthermore, there is a lack of data to confirm that SNAP23 is not essential for photoreceptor function, as it's likely that its absence can be compensated for by SNAP25 or other isoforms.

- As the reviewer suggested, SNAP-23 may not be able to fill out sub-millisecond exocytosis, however, SNAP-23 still has the potential to regulate exocytotic functions. For example, SNAP-23 knockout was shown to enhance the release of insulin from pancreatic beta cells (Liang et. al., 2020), while SNAP-23 overexpression inhibits insulin secretion (Chen et. al., 2023). Presumably because SNAP-23 competes with SNAP-25 for exocytosis and SNAP-23 is less efficient than SNAP-25, it is possible that SNAP-23 removal could enhance ERGs responses. However, we did not see such effects. Nevertheless, as the reviewer pointed out, we cannot completely eliminate the fact that SNAP-25 could compensate for the lack of SNAP-23, and we include these points in our discussion (lines 356-364).

4. The authors have made only a limited effort to establish connections between the synapses of photoreceptors and auditory sensory cells. Notably, they have overlooked the recent work by Calvet et al. (PMID 36483015), which offers fresh insights into the function of SNAP25 in these cells. This omission is unfortunate because it could have enriched both results and discussion within the current article. In the realm of research, keeping updated to date is not just important but necessary to progress.

-We agree Calvet et al. is a significant article that provides insights onto SNAP-25 functions in the IHC of the auditory system. We have now added citations to it in our discussion to enhance the connections between the sensory receptors in visual and auditory systems (lines 306-312).

Minor comments

1. The clarity of the writing, as well as the quality of the figures and their legends, can be enhanced to better facilitate the reader's understanding. Each immunofluorescence panel should incorporate high-magnification views to emphasize the authors' assertions.

- We have revised our figures with more labels and updated figure legends with more information (sample size, retinal region represented, statistics information, etc.). We have also gone in and revised our text to increase clarity according to referee suggestions.

2. it would be helpful to mention the retinal region in all figures.

- We have now stated the region selected for imaging in all represented figures.

REVIEWERS' COMMENTS:

Reviewer #1 (Remarks to the Author):

The revised version of the manuscript presented by Huang and colleagues on the roles of SNAP25 in mouse photoreceptors fully addresses my scientific concerns.

The manuscript would still benefit from being edited for clarity, synthesis of information, and grammar, particularly in some of the sections that have undergone revision. In addition, there are a few sentences that should be modified (see below).

Comments

1. Line 165 – it is difficult to conclude that SNAP-23 is functionally dispensible unless one knows that there has been no compensation. Given that the Cre turns on so early in development, it could be that there is compensation by another SNAP member. Furthermore, the ERG's are not likely to be sufficiently sensitive to detect subtle roles for SNAP23. They authors need to qualify this sentence => they could say "seems" to be functionally dispensible.

2. Line 200 – Why do the authors state that the "second order" retinal cells degenerate? Do the authors mean first order (i.e. photoreceptors)?

3. Lines 253– strictly speaking, these are examples of bipolar cell dendritic sprouting. It would be more accurate to refer to them as the sprouting of dendrites normally postsynaptic to photoreceptors, rather than as ectopic synapses, since sprouting can occur after photoreceptor loss and as such, there is no synapse.

4. Lines 303 and 338–Several studies have shown that the major syntaxin 3 expressed by mammalian photoreceptors is syntaxin 3B and not the ubiquitously expressed syntaxin 3 known as syntaxin 3A. Please revise for accuracy. One can say simply "syntaxin 3" but one should not incorrectly state that it is the "ubiquitously expressed syntaxin 3."

5. Lines 353-355. The meaning of this section is unclear.

Reviewer #2 (Remarks to the Author):

The authors have revised the manuscript according to the reviewers suggestions. The points of criticism raised by the reviewer have been addressed in the revised version of the manuscript. I recommend publication of the revised manuscript in its present form.

Reviewer #3 (Remarks to the Author):

Having thoroughly examined the revised manuscript by Huang and colleagues, I found notable improvement. The authors adequately addressed most of my concerns. There is evident effort in refining the econography, aligning, and updating the discussed literature.

Reviewer #1 (Remarks to the Author):

The revised version of the manuscript presented by Huang and colleagues on the roles of SNAP25 in mouse photoreceptors fully addresses my scientific concerns.

The manuscript would still benefit from being edited for clarity, synthesis of information, and grammar, particularly in some of the sections that have undergone revision. In addition, there are a few sentences that should be modified (see below).

- We thank the reviewer for carefully reading our revised manuscript. We are pleased that the reviewer found the previous concerns to be sufficiently addressed. We have also revised our manuscript again according to the reviewer's suggestions to increase clarity.

Comments

1. Line 165 – it is difficult to conclude that SNAP-23 is functionally dispensable unless one knows that there has been no compensation. Given that the Cre turns on so early in development, it could be that there is compensation by another SNAP member. Furthermore, the ERG's are not likely to be sufficiently sensitive to detect subtle roles for SNAP23. They authors need to qualify this sentence => they could say “seems” to be functionally dispensable.

- We have revised this sentence to “Overall, weakly expressed *SNAP-23* in photoreceptor cells seems to be functionally dispensable.” (line 166-167).

2. Line 200 – Why do the authors state that the “second order” retinal cells degenerate? Do the authors mean first order (i.e. photoreceptors)?

- We originally meant that after photoreceptors generated, we observed a secondary degeneration of the inner retina too (Figure 4A, decreased INL and IPL in the SNAP-25 cKO retina representative). As such, we changed “second order retinal” to “inner retinal” and defined the cells we meant in the revised manuscript (lines 203-204). We hope this clears up the ambiguity.

3. Lines 253– strictly speaking, these are examples of bipolar cell dendritic sprouting. It would be more accurate to refer to them as the sprouting of dendrites normally postsynaptic to photoreceptors, rather than as ectopic synapses, since sprouting can occur after photoreceptor loss and as such, there is no synapse.

- We have now revised this statement for accuracy to “The postsynaptic region also showed distortions at P9 in SNAP-25 cKO animals (Fig. 7B, 7C), which we attribute to the sprouting of dendrites normally postsynaptic to photoreceptors,” (line 259).

4. Lines 303 and 338–Several studies have shown that the major syntaxin 3 expressed by mammalian photoreceptors is syntaxin 3B and not the ubiquitously expressed syntaxin 3 known as syntaxin 3A. Please revise for accuracy. One can say simply “syntaxin 3” but one should not incorrectly state that it is the “ubiquitously expressed syntaxin 3.”

- We have now removed the words “ubiquitously expressed” from these sentences in our revised manuscript (lines 308 and 343).

5. Lines 353-355. The meaning of this section is unclear.

- We have specified what we meant by regulatory effects and fixed the previous run-on sentence (lines 360-361). We hope this makes the section clearer.

Reviewer #2 (Remarks to the Author):

The authors have revised the manuscript according to the reviewers suggestions. The points of criticism raised by the reviewer have been addressed in the revised version of the manuscript. I recommend publication of the revised manuscript in its present form.

- We thank the reviewer for taking their time and reading our revised manuscript. We are happy the reviewer recommends the publication of our manuscript.

Reviewer #3 (Remarks to the Author):

Having thoroughly examined the revised manuscript by Huang and colleagues, I found notable improvement. The authors adequately addressed most of my concerns. There is evident effort in refining the econography, aligning, and updating the discussed literature.

- We thank the reviewer for thoroughly going through our revised manuscript and are pleased that the reviewer recognizes our effort and improvements.

With this, we believe the remaining concerns to be addressed.